

# The UKC3 regional coupled environmental prediction system

Huw W. Lewis[1], Juan Manuel Castillo Sanchez[1], Alex Arnold[1], Joachim Fallmann[1,a], Andrew Saulter[1], Jennifer Graham[1,b], Mike Bush[1], John Siddorn[1], Tamzin Palmer[1], Adrian Lock[1], John Edwards[1], Lucy Bricheno[2], Alberto Martínez de la Torre[3], James Clark[4]

[1]Met Office, Exeter, EX1 3PB, UK
[2]National Oceanography Centre, Liverpool, L3 5DA, UK
[3]Centre for Ecology & Hydrology, Wallingford, OX10 8BB, UK
[4]Plymouth Marine Laboratory, Plymouth, PL1 2LP, UK
[a]now at: Institut für Physik der Atmosphäre, Johannes Gutenberg-Universität Mainz, Germany
[b]now at: Centre for Environment, Fisheries and Aquaculture Science, Pakefield Rd, Lowestoft NR33 0HT, UK

*Correspondence to*: Huw W. Lewis (huw.lewis@metoffice.gov.uk)

**Abstract.** This paper describes an updated configuration of the regional coupled research system, termed UKC3, developed and evaluated under the UK Environmental Prediction collaboration. This represents a further step towards a vision of simulating the numerous interactions and feedbacks between different physical and biogeochemical components of the environment across sky, sea and land using more integrated regional coupled prediction systems at km-scale resolution. The UKC3 coupled system incorporates models of the atmosphere (Met Office Unified Model), land surface with river routing (JULES), shelf-sea ocean (NEMO) and ocean surface waves (WAVEWATCH III), coupled together using OASIS3-MCT libraries. The major update introduced since the UKC2 configuration is an explicit representation of wave processes in the ocean and their feedbacks through wave-to-ocean coupling. Ocean model results demonstrate that wave coupling, in particular representing the wave modified surface drag, has a small but positive improvement on the agreement between simulated sea surface temperatures and in situ observations, relative to simulations without wave feedbacks. Other incremental developments to the coupled modelling capability introduced since the UKC2 configuration are also detailed.

Coupled regional prediction systems are of interest for applications across a range of timescales, from hours to decades ahead. The first results of simulations run over extended periods, covering four experiments each of order one month in duration are therefore analysed and discussed in the context of further characterising the potential benefits of coupled prediction on forecast skill, and on the stability of such systems over longer time periods. Results across atmosphere, ocean and wave components are shown to be of at least comparable skill to the equivalent uncoupled control simulations, with notable improvements demonstrated in surface temperature and wave state predictions in some near-coastal regions, and in wind speeds over the sea.



## 1 Introduction

This paper describes the third release of a regional coupled prediction system, termed UKC3, developed to support research to improve the simulation and understanding of the various interactions and feedbacks between different physical and biogeochemical components of the atmosphere, ocean and land across the UK and north-west European shelf region. The
UKC3 system represents an incremental update to the second research-mode system, UKC2, described by Lewis et al. (2018). This paper provides a description of the enhancements to model components and of new coupling science introduced within the latest configuration, and reports on system performance based on new simulations over longer evaluation periods than used to describe the UKC2 performance.

### 1.1 Motivations for regional coupled model development

Coupled Earth system modelling on global scales, encompassing representation of the physical and biogeochemical feedbacks and interactions between the atmosphere, oceans, cryosphere and land surface is a well established and mature science discipline, particularly in the context of longer timescale applications from seasonal-range forecasting out to climate change prediction. For applications on shorter timescales or requiring information at more localised scales, including weather forecasting, regional climate scenarios, land management and marine forecasting for example, a discipline of regional coupled
prediction has evolved over recent years. This is motivated by a drive from both research and operational applications to develop more wholistic simulations of the environment at high resolution in which the numerous Earth System feedback processes are more explicitly represented (e.g. Shapiro et al., 2010; Pullen et al., 2017a). These systems enable improved understanding of how heat, momentum, freshwater and biogeochemical exchanges affect both marine and atmosphere-land systems.

A number of complex interactions and feedbacks between air, sea and land only become relevant when considering the environment at more localised scales of order kilometres. At these scales for example, mesoscale features such as ocean eddies begin to dominate air-sea interaction processes (e.g. Frenger et al., 2013; Byrne et al., 2015; Oerder et al., 2018), the local landscape and details of precipitation processes become relevant for linking meteorology with catchment-scale hydrology (e.g. Kay et al., 2015; Clark et al., 2016; Kendon et al., 2017), and the influence of freshwater flows on the coastal marine
environment become apparent (e.g. Simpson, 1992; Dzwonkowski et al., 2017).

The coastal zone is particularly critical in this context – where the feedbacks between atmosphere, land and ocean state all interplay, and where significant populations live and critically important national infrastructure are sited. The impacts of feedbacks are often manifested through natural hazards, including coastal inundation, flooding and erosion resulting from high waves and storm surge or development of harmful algal blooms and impacts on aquaculture for example. Typically natural
hazards from multiple sources may combine or occur concurrently (e.g. Forzieri et al., 2016; Lewis et al., 2015). It is therefore hypothesised that the predictive skill across atmosphere, land hydrology, ocean and wave systems can be improved through explicitly representing the feedbacks between them. Provision of information from coupled systems might also enable an



improvement in the range and consistency of actionable information that can be provided through hazard warnings and guidance.

These drivers equally apply on longer timescales, over which the impact of feedbacks on the mean state and extremes may be more significant, and a full Earth system approach may prove to be beneficial for developing relevant regional-scale
information of use for planning and policy-making applications (e.g. Miller et al., 2017).

Finally, km-scale regional prediction tools applied in different regions around the world provide a testbed to inform parameterization development in coarser scale systems. As the availability and processing power of high-performance computing increases allowing more routine high-resolution application of global-scale atmosphere (e.g Jung et al., 2012; Walters et al., 2017), hydrology (e.g. Bierkens et al., 2015; Emerton et al., 2016), ocean (e.g. Hewitt et al., 2017; Holt et al.,
2017) and even Earth System (e.g. Palmer, 2012) prediction systems, developing effective coupling mechanisms at km-scale becomes increasingly relevant.

## 1.2 Recent progress in regional coupled model research, development and application

The km-scale regional coupled prediction approach is already beginning to reach operational maturity in some forecasting centres and for specific contexts. For example, Durnford et al., (2018) describe the implementation of an integrated water cycle
prediction system for the Great Lakes and St Lawrence river by Environment Canada, serving a range of applications for industries and populations with exposure to lake water levels. Kunii et al. (2017) and Wada and Kunii (2017) discuss the development of a strongly coupled regional atmosphere-ocean data assimilation system with the Japan Meteorological Agency's configurations, and its potential to improve tropical cyclone prediction through improved representation of the sea surface temperature (SST) initial condition and evolution.

The underpinning research required to improve regional coupled prediction systems, and their application to support process-based research also continues. This is supported by learning from ongoing development of coupling parameterisations and their application in global-scale coupled systems (e.g. Mogensen et al., 2017; Shimura et al., 2017; Hirons et al., 2018; Donelan et al., 2018). There is also a critical dependence on the ongoing collection and analysis of relevant air-sea flux measurements in different regions for improving process understanding and supporting model evaluation (e.g. Hackerott et al., 2018;
Vinayachandran et al., 2018).

Research using a range of km-scale regional coupled prediction systems continues to deliver new insights. Developing increased understanding and system improvements benefit from the application of a diversity of component model and coupling technologies in a range of environments.

For example, Wahle et al. (2017) demonstrated complimentary improvements in both wave and wind forecasts in the complex
coastal region of the southern North Sea by implementing wave-induced drag computed by the WAM wave model (Komen et al., 1994) running at 5 km resolution in the COSMO regional atmosphere model (Rockel et al., 2008) run at 10 km resolution. Two-way coupling was achieved using the OASIS3-MCT coupler (Valcke et al., 2015) every 3 minutes during the simulations.





Both significant wave height and wind speeds were reduced by order 8% and 3% respectively over a 3-month mean due to the extraction of energy and momentum from the atmosphere by waves. Gronholz et al. (2017) studied the impact of ocean-atmosphere interactions on ocean stratification over a similar model domain. Results demonstrated both the sensitivity of SST to the resolution of atmospheric forcing and that enhanced vertical mixing in the fully coupled ocean simulation during a storm

event could have potential impacts for prediction of phytoplankton bloom development. This study applied the Coupled Ocean-Atmosphere-Wave-Sediment Tranport (COAWST; Warner et al., 2010) system, based on the WRF (Weather Research & Forecasting; Skamarock et al., 2008) atmosphere model coupled using the Model Coupling Toolkit (MCT; e.g. Larson et al., 2005) to the ROMS (Regional Ocean Modeling System; Shchepetkin and McWilliams, 2005) ocean model, both run at around 10 km horizontal resolution and with a 10 minute coupling frequency between components.

Ricchi et al. (2017) also applied the COAWST system to demonstrate the sensitivity of a Tropical-Like Cyclone case study in the Mediterranean Sea (often termed 'Medicanes') to coupling. This study also implemented coupling between the atmosphere and ocean to the SWAN (Simulating WAves in Nearshore; Booij et al., 1999) wave model. The system was applied at a 5 km horizontal resolution, with model fields exchanged between components every 5 minutes. While coupling was found to improve the simulation of heat and momentum fluxes for example, it was also highlighted that the sensitivity to details such

as the surface roughness parameterization used was greater than the sensitivity to coupling. The beneficial impact of improved SST initial condition and its dynamic evolution through coupling on the simulation of heavy rainfall events in the Mediterranean was discussed by Rainaud et al. (2017), who applied a coupled simulation of the AROME atmosphere (2.5 km resolution) and NEMO ocean (1/36° resolution) models.

Atmosphere-wave coupling over the Mediterranean during a cyclonic event was also assessed by Varlas et al. (2017) applying

two-way coupling between WRF atmosphere (10 km resolution) and WAM wave models using the OASIS3-MCT coupler. Coupling was found to impact the evolution of the system, with similar reductions in wind speed and wave height to that discussed by Wahle et al. (2017), and result in an overall improvement of forecast wave height skill by up to 20% and wind speed by up to 5% over the sea. This is ultimately anticipated to lead to improved operational warnings and guidance to users. Recent research focussed on other locations includes the work of Pullen et al. (2017b), who successfully applied a regional

coupled model to assess the role of air-sea feedbacks on vortex shedding in the lee of Maderia Island. This modelling system incorporated nested implementations of the NOGAPS atmosphere and NCOM ocean models, run at up to 2 km horizontal resolution and with coupled fields exchanged every 6 minutes using the Earth System Modeling Framework (ESMF) coupler through the 20 day simulation period. Seo et al. (2017) used the Scripps Coupled Ocean-Atmosphere Regional model (SCOAR; Seo et al., 2007) over the Arabian Sea, in which WRF and ROMS models were run on 9 km resolution grids and coupled every

6 hours. In addition to demonstrating local influences of SST-wind and current-wind interations in the region, Seo et al. (2017) noted the potential downstream influence and adjustment of the monsoon circulation due to air-sea interaction. Oerder et al. (2018) illustrated the impact of including ocean surface currents in the calculation of atmospheric wind stress, in particular above regions with coherent ocean eddies, in a study region around the eastern Pacific Ocean, Peru and Chile. This research



applied a 1/12° resolution implementation of the WRF atmosphere model coupled to a NEMO ocean model on the same horizontal grid, coupled at an hourly frequency using the OASIS3-MCT coupler library. Regional model coupling was also found to improve the simulation of extreme rainfall over Brazil by Luiz do Vale Silva et al. (2018), who applied COAWST at 12 km resolution and found intensification of rain-bearing systems driven by warm SST across the Atlantic Ocean off the coast

of Brazil.

A number of other studies continue to examine atmosphere-land-ocean feedback processess in very near coastal estuarine environments. For example, Marsooli and Lin (2018) applied a two-way coupled ocean-wave prediction system in the New York – New Jersey region for a simulation of Hurricane Sandy to illustrate the benefit of representing coupled feedbacks in an extreme event for improving storm tides. Akan et al. (2017) applied a nested implementation of the COAWST coupled

modelling system to examine wave-current interactions at the mouth of the Columbia River. Results show an asymmetric impact of current-induced modification to the wave field, with waves amplified at the mouth of the river due to the impact of tides. The effect of tidal, wind and wave forcing in a near-coastal environment was also highlighted through detailed observations of the Rhine river region of freshwater influence by Flores et al. (2017).

## 1.3 Evolution of the UK and north-west shelf regional coupled system

Lewis et al. (2018) detailed the rationale for developing regional coupled prediction capability at km-scale resolution for a UK and north-west shelf focused domain, and described the underpinning atmosphere and ocean boundary layer exchanges of momentum and heat, and of the fluxes of freshwater between the atmosphere and land systems before entering the ocean as river discharge. The UKC2 evaluation framework used to run and understand the impact of coupling in case study simulations was also described. This approach and associated naming conventions continues to support the research activities associated

with evaluating UKC3 discussed in this paper. Results of the case studies described by Lewis et al. (2018) demonstrated that model performance could be achieved with the UKC2 system that was at least of comparable skill to its component control simulations, with examples where improvements in agreement against in situ observations could be achieved for atmosphere, ocean and wave variables assessed. Further research relevant to the UKC2 system is also described by Fallmann et al., (2017) and Martinez et al. (2018).

A number of limitations and priorities for short and longer-term future development were also identified by Lewis et al. (2018). The following specific aspects have been addressed within the UKC3 configuration development and are discussed further in this paper.

1.  Improving the functionality and flexibility of use of the coupled prediction system (see Sect. 2),
2.  Development of wave-to-ocean coupling physics (see Sect. 3),

3.  Revisiting a number of assumptions and parameterizations embedded within component models,   (see Sect. 3)
4.  Performing longer simulations, expanding on an initial series of 5-day case studies (see Sect. 4),



This paper is organised as follows. Section 2 introduces the UKC3 regional coupled prediction system, providing details of updates to the UKA3 atmosphere, UKL3 land surface, UKO3 ocean and UKW3 wave model configurations since the preceding configurations for each component described by Lewis et al. (2018). Section 3 describes the wave-to-ocean coupling physics introduced within UKC3 configurations. Results from new simulations using the UKC3 configurations during four contrasting

month-long experiments are presented in Sect. 4. Conclusions and priorities for future development are discussed in Sect. 5.

## 2 The UKC3 regional coupled prediction system

The third release of the regional coupled prediction system UKC3 consists of configurations of the Met Office Unified Model (MetUM) atmosphere (version 10.6; e.g. Brown et al., 2012), and JULES (Joint UK Land Environment Simulator) land surface model (version 4.7; Best et al., 2011; Clark et al., 2011), coupled to the NEMO (Nucleus for European Models of the Ocean)

model (version3.6, revision 6232; Madec et al., 2016) and WAVEWATCH III wave model (version 4.18; Tolman et al., 2014). Coupling is achieved through use of the OASIS3-MCT (Ocean-Atmosphere-Sea Ice-Soil) coupling libraries (version 2.0; Valcke et al., 2015). A naming convention is adopted whereby the atmosphere-land (MetUM-JULES) configuration is termed UKA3, and similarly the uncoupled ocean and wave components as UKO3 and UKW3 respectively.

Table 1 provides a summary of the key differences and similarities between the UKC3 system and the previous UKC2

configuration as described by Lewis et al. (2018). The update of atmosphere, land surface, ocean and wave model codes used in UKC3 in itself represents the addition of new science, inherited from underpinning development of the component model science. Only those aspects where the model codes used in UKC3 have substantially developed between configurations are highlighted in the following sections. The model domain and grid definitions are identical to the UKC2 configuration. The extent of the UKC3 system domain is illustrated in Fig. 1, together with an illustration of the available in-situ observing

networks used for model evaluation.

The overall approach to system development and the framework for running simulations as rose suites is as described by Lewis et al. (2018). Table 2 summarises the different coupled and uncoupled configurations defined as part of the UKC3 research system. This also introduces the naming convention adopted in order to run the same science and coupling configuration but with different initial conditions or external forcing. Where more than one option is available for a particular configuration, the

required configuration can be specified by setting a RUNID environment variable prior to running a simulation.

A number of terms describing each simulation approach are introduced, as follows.

> *Fully coupled:*
>
> > two-way feedbacks represented between all model components within the system.
>
> *Partially coupled:*

> > two-way feedbacks represented between only two components of the system. In the ocean-wave coupled
> > UKC3owg configuration for example, atmospheric forcing is provided from the external operational global





MetUM archive, although with wave-modified surface drag coefficient used in the calculation of atmospheric stress from wind components (see Sect. 3.1).

*Forced mode:*

information is provided on the state of external components (e.g. the wave state in the ocean model – UKO3gw; or the ocean state in a wave model – UKW3go) as updating surface boundary conditions via file forcing, with no feedbacks represented of the effect of either component on each other. Note that forced mode results are not discussed further in this paper for simplicity.

*Uncoupled (control):*

default mode simulations for a given model component, in which no feedbacks with external components are represented. For the UKA3u atmosphere-only control simulations, the SST lower boundary condition is updated with OSTIA data each day and kept constant throughout the day, surface ocean currents are assumed to be zero and a default Charnock parameter constant of 0.011 is assumed. This is in contrast to the UKA3g or UKA3h configurations for which the initial condition SST would be persisted for the entire duration of a simulation, as applied by Lewis et al. (2018) for 5-day duration case study tests. For the UKO3g ocean-only control simulations, only hourly wind forcing and three-hourly radiation and moisture fluxes are applied, read as external files from the operational global-scale MetUM archive. For the UKW3g wave-only control simulations, only hourly wind forcing is applied, read as external files from the operational global-scale MetUM archive.

Namelists describing the configuration for all components discussed in this paper are defined as suites under the rose framework for managing and running model systems (http://metomi.github.io/rose/doc/rose.html). All configurations described are made available as rose suites to registered researchers under a repository at https://code.metoffice.gov.uk/trac/roses-u. A more detailed description of the namelists used across all configurations is also included in the Supplementary Material to this paper.

Table 3 lists the coupling exchanges of model variables between each component within UKC3. A total of 24 variables are exchanged, with 6 new exchanges introduced between the WAVEWATCH III wave and NEMO ocean models in UKC3 to support representation of wave-to-ocean feedbacks (discussed further in Sect. 3). Castillo et al. (2017) considered a number of aspects related to the optimisation and computational costs of the UKC2 system, and assessed that the system run times are largely insensitive to the number of fields exchanged between components.

## 2.1 The UKA3 atmosphere component

The atmosphere model component within UKC3 (named UKA3 when run in atmosphere-only mode) uses the RA1-M regional atmosphere science configuration described in detail by Bush et al. (2018). This is implemented using the MetUM code at version 10.6 (e.g. Walters et al., 2017). Table 4 highlights the key similarities and differences between UKA3 and UKA2





configurations. Updating between MetUM model code vn10.1 and vn10.6 introduces a substantial number of incremental scientific and technical fixes, enhancements and optimisations, delivered through ongoing model evaluation and development.. Technical details on the RA1 science configuration are also provided to registered users of the MetUM code at https://code.metoffice.gov.uk/trac/rmed/wiki/ra1. Key changes introduced between UKA3 (RA1) and the UKA2 science

configuration in model code or namelist options used of relevance to the regional atmosphere performance are highlighted below. The corresponding namelist changes are listed in Table 5.

**Improvements to simulation of low cloud and fog processes (RA1 ticket #1)**

Following comparison of 1.5 km resolution model data to field campaign observations (Boutle et al., 2018), changes have been applied to the prescription of cloud droplet number variation with height. Cloud droplet numbers are set to a fixed parameter

*ndrop_surf* below a defined height *z_surf* above the surface (see Table 5). While based on observations over land, this change is considered important in the context of the UK regional coupled prediction system development given the strong sensitivity of fog and near surface cloud to air-sea coupling (e.g. Fallmann et al., 2017; Fallmann et al., 2018).

**Improvement to simulation of convective precipitation through moisture conservation (RA1 ticket #2)**

Long-term application and evaluation of convective-scale configurations of the MetUM (e.g. Clark et al., 2016) have shown

that the semi-Lagrangian advection scheme can give rise to spurious sources of moisture, and notably excessive rainfall rates, in the vicinity of resolved convection. The RA1-M configuration introduces global conservation of moisture species following Aranami et al. (2015) (defined by updated *run_dyn* and *run_sl* namelist parameters in Table 5). This has a significant beneficial impact on the mean rainfall rates in convective situations, reducing the domain mean accumulations and removing unrealistic extreme precipitation rates (Bush et al., 2018). This improvement is particularly important in the context of fully coupled

environmental predictions and a vision of a more integrated representation of the hydrological cycle across atmosphere, land and ocean (e.g. Lewis et al., 2018).

**Atmospheric boundary layer simulation enhancements (RA1 tickets #5, #10, #12, #15)**

A number of incremental updates have been introduced in the RA1-M science configuration, and thereby in UKA3, related to the atmospheric boundary layer paramterization (*run_bl* parameters in Table 5). Entrainment fluxes are now defined across a

diagnosed inversion thickness at the boundary layer top rather than a previously assumed sharp sub-grid layer. Further details on this and other more incremental boundary layer mixing scheme updates are provided by Walters et al. (2017; see discussion of GA7 ticket #83) and Bush et al. (2018).



**Improved atmospheric absorption and surface radiative fluxes in the MetUM radiation scheme (RA1 ticket #9)**

The RA1-M science configuration adopts the same treatment of gaseous absorption as used in the GA7 MetUM configuration, described by Walters et al. (2017; see discussion of GA ticket #16). Briefly, an updated solar spectrum is used for short-wave radiation and improvements are made to the representation of atmospheric composition. These changes result in increased absorption and reduced surface short-wave fluxes in clear-sky conditions. At longwave bands, clear sky outgoing longwave radiation is reduced and the downwards surface flux increased relative to the UKA2 definition.

**Time-correlated stochastic PBL perturbations to improve triggering of showers (RA1 ticket #25)**

The effectiveness of the stochastic boundary layer perturbations, applied in the vicinity of cumulus clouds only, at triggering resolved scale convection has been improved in RA1-M (configured with the *run_stochastic* namelist options listed in Table 5). Random heating increments can persist for several minutes, and both temperature and moisture perturbations are now applied. Perturbations are based on the surface buoyancy flux, with a maximum possible value (*mag_pert_theta*) of 1.0 K and an option enabled in RA1-M (*l_pert_shape=.true.*) is also included to weight the perturbations more strongly to the middle of the boundary layer and not at all at the surface. Perturbations are applied in the RA1-M configuration up to a maximum height *z_pert_theta* of 1500 m.

**2.1.1 Modifications to MetUM for regional atmosphere coupling**

As for UKC2, exactly the same codes are compiled and built for both coupled and uncoupled cofigurations to ensure all simulations are run with identically built code. A number of required code adaptations were implemented as branches to the vn10.6 MetUM trunk code. These are detailed for reference in the Supplementary Material. In general, these modifications can be categorised as being required either to:

- Apply the RA1-M graupel definition in the JULES snow scheme (Sect 2.2),
- Couple effectively between ocean and atmosphere grids with mismatching coastlines, due to grid interpolation of mis-matched land/sea masks (as in Lewis et al., 2018),
- Enable dynamic coupling and exchange of information between the atmosphere and a wave model,
- Enable river routing within the coupled MetUM-JULES system,
- Enable consistent coupling of snow when convective snow is explicitly resolved.

To encourage collaboration, a single merged copy of the UKA3 MetUM model code is available to registered researchers via a shared code repository, which can be accessed via https://code.metoffice.gov.uk/trac/utils/browser/ukeputils/trunk/gmd-2018. The code repository location is also linked directly for registered researchers in Table 4.





## 2.2 The UKL3 land surface component

The JULES land surface component within UKC3 is implicitly coupled to the MetUM atmosphere model code, using the method of Best et al. (2004), in all configurations with an atmosphere component. Table 6 lists the key similarities and differences between the land surface specification in UKA3 and UKA2. Details of the trunk code updates between JULES

release vn4.2 and vn4.7 can be accessed at http://jules-lsm.github.io/vn4.7/. The majority of changes however are not considered relevant for the regional land surface component. The UKL3 definition is also a direct implementation of the land surface settings associated with the RA1-M science configuration, with additional options and parameters enabled for river routing. Key changes introduced in UKL3 are highlighted below, with corresponding namelist changes given in Table 7.

### Improved representation of land surface properties to improve near surface temperature biases (RA1 ticket #3)

Four related updates have been implemented in an attempt to reduce clear-sky surface temperature biases over land. These include reducing the amount of bare soil defined and changing the scalar roughness and albedo of vegetated tiles. An updated land use ancillary of land surface tile fractions was generated, using the CCI land cover data set (CCI, 2018) for parts of the domain away from the UK, and taking greater care to account for seasonal variations of the bare soil fraction as a function of the leaf area index (LAI). Further discussion is provided by Walters et al. (2017; see GA ticket #30) and Bush et al. (2018).

Further, scalar roughness length parameters over grass tiles were reduced, by reducing its ratio to the momentum roughness from 0.1 to 0.01 (Table 7). This enhances the difference between surface and air temperatures, in closer agreement with field observational studies. The JULES albedo parameters *alnir_io*, *alpar_io*, *omega_io* and *omnir_io* were also revised (Table 7).

### Ignoring graupel in treatment of JULES snow surfaces (RA1 ticket #19)

The JULES namelist parameter *graupel_options* is set to 1 in RA1-M to avoid the default behaviour of JULES including

graupel as snow in the surface scheme when graupel is included in the MetUM surface snowfall diagnostic. While this maintains conservation of water and energy, the properties assigned to new snowfall are considered inappropriate for graupel and this can degrade the surface evolution.

### Updated land surface hydrology parameters and runoff generation algorithm

The overall vision and initial implementation for representing the hydrological cycle across UKC2 coupled components was

introduced by Lewis et al. (2018). The UKC3 system adopts the same configuration, which are not part of the standard RA1 definition. Martinez et al. (2018) provide a detailed description of the numerous offline tests and conclusions drawn for optimising the JULES hydrology parameters in order to generate improved runoff characteristics and agreement between river flow simulations and gauge observations.



### 2.2.1 Modifications to JULES for regional coupling

Similar to the UKC2 system, a number of required code adaptations were implemented as branches to the vn4.7 JULES trunk code to enable regional coupling and run the MetUM-JULES coupled system with river routing. These are detailed for reference in the Supplementary Material. In general, these modifications can be categorised as being required either to:

- Apply the RA1 graupel definition in the JULES snow scheme,
- Enable dynamic coupling and exchange of information between the atmosphere and a wave model,
- Apply the Martinez et al. (2018) approach of a slope-dependent Probability Distribution Model runoff generation,
- Enable river routing within the coupled MetUM-JULES system,
- Apply a check in the calculation of surface exchange coefficients for slightly unstable conditions.

A single merged copy of the UKL3 JULES model code is also made available to registered researchers via a shared code repository, which can be accessed via https://code.metoffice.gov.uk/trac/utils/browser/ukeputils/trunk/gmd-2018. The repository location is also linked in Table 7.

### 2.3 The UKO3 ocean component

The most significant change related to coupling introduced between UKC2 and UKC3 is the implementation and configuration
of wave-to-ocean feedbacks within the NEMO ocean model code. Further details are provided in Sect. 3.

Table 8 highlights other common and differing aspects of the UKO3 regional shelf-seas ocean-only configuration relative to UKO2. Updates were introduced in order to maintain a common science configuration to the evolving Atlantic Margin Model (AMM15) ocean only shelf-seas forecasting system, which is described in detail by Graham et al. (2018). This required updating the NEMO vn3.6 trunk code revision from r5518 to r6232, which includes a number of minor bug fixes and technical
code improvements only. The following configuration changes were also implemented.

#### Solar radiation penetration and surface restoring parameters

Arnold (2018) describes a number of sensitivity tests conducted using the AMM15 regional ocean model configuration to investigate the impact of different choices in the specification of surface meteorological forcing on simulated SST. Particular consideration was given to the appropriate choice for the ratio of penetrating to non-penetrating shortwave solar radiation in
the NEMO "RGB" light penetration scheme (Madec et al, 2016). This study confirms that improved summer time SST were produced when using a *rn_abs* ratio of 0.66, indicating 66% absorption at the surface.

Arnold (2018) also highlight the importance of using a surface restoration scheme (*ln_ssr=.true.*, *ln_ukmo_haney=.true.*) for ocean-only simulations using UKO3. This scheme nudges the simulated SST towards OSTIA, in order to correct for discrepancies between the surface temperatures consistent with the atmospheric forcing and the evolving ocean model
climatology. Note that surface restoring was not implemented in UKO2 configurations. It is also not appropriate to apply these



corrections when running in ocean-atmosphere coupled mode, given that the atmospheric fluxes are consistent with the underlying ocean model by definition.

**Updated Baltic Sea boundary condition**

For simplicity in UKO2, the inflow to the domain from the Baltic Sea at the eastern boundary was treated as two river sources,

located in the Kattegat strait. In UKO3, as in AMM15, eastern boundary conditions are instead taken from a regional Baltic simulation (Gräwe et al., 2015). Baltic boundary conditions are applied over a relaxation zone of horizontal width (*nn_rimwidth*) 10 grid cells, while boundary conditions into the majority of the domain along the remaining edges are applied over a relaxation zone of 15 grid cells.

**River outflows**

Pending further testing and more thorough evaluation of the integrated atmosphere-land system for simulating river flows, by default the river runoff fluxes applied at ocean model coastal grid cells uses a climatology as described by Graham et al. (2018), rather than applying the MetUM-JULES calculated flows.

**2.3.1 Modifications to NEMO for regional coupling**

Sect. 3 provides a more detailed discussion of the implementation of wave-to-ocean coupling within UKC3, which required a

number of code modifications and changes to namelist parameter settings beyond the AMM15 configuration. A single merged copy of the UKO3 NEMO model code has been prepared to be available to registered researchers via a shared code repository, which can be accessed via https://code.metoffice.gov.uk/trac/utils/browser/ukeputils/trunk/gmd-2018. The exact location is linked in Table 8. In general, modifications to the NEMO vn3.6 r6232 trunk code are made to:

- apply capability specific to running NEMO for a domain including a shelf-seas region
- implement wave to ocean coupling physics
- enable NEMO to run within a coupled system without using the MetUM coupling utilities
- ensure physically sensible coupled data exchanges in regions of unaligned atmosphere and ocean land/sea masks
- when enabled, apply river flux coupling within a sub-domain (UK coastlines only) of the UKC3 domain

**2.4 The UKW2 surface wave component**

Table 9 reflects the close similarity in configuration between UKW2 and UKW3 wave model components. This is a result of model code developments being limited to the provision of new coupled fields to support wave-to-ocean coupling (Sect. 3) and some minor bug fixes. A copy of the WAVEWATCH III model code used to define the UKW3 configuration is made available via https://code.metoffice.gov.uk/trac/utils/browser/ukeputils/trunk/gmd-2018 to researchers who are registered as WAVEWATCH III users. The exact path is also linked from Table 9.



## 3 Representing wave-ocean interactions

A key gap in the UKC2 regional coupled configuration requiring further system development identified by Lewis et al. (2018) was the lack of representation of wave-to-ocean coupling physics. In addition to the feedbacks to the overlying atmosphere through modifying surface roughness, it is well known that surface waves modify momentum exchanges and mixing in the

underlying ocean surface boundary layer through a number of different processes. The main interactions represented in the fully coupled atmosphere-ocean-wave UKC3aow system are:

> i) the modification of surface stress by wave growth and dissipation,
>
> ii) Stokes-Coriolis force,
>
> iii) wave height dependent ocean surface roughness.

When forced by an uncoupled atmosphere, the effect of waves in modifying the atmospheric boundary layer can also be accounted for in modifying the calculation of surface stress from wind speed in the NEMO surface forcing. Note that modifications to the NEMO turbulent kinetic energy (TKE) budget due to wave processes are not included in UKC3.

Breivik et al. (2015) set out the physical basis for the representation of surface wave effects in the NEMO ocean model, as implemented in the global coupled forecast system at the European Centre for Medium-Range Weather Forecasting (ECMWF).

Results demonstrate reduced sea surface and sub-surface temperature biases and improvements in the simulated total ocean heat content relative to observations when wave effects are included. On regional scales of relevance to the UKC3 system, a number of previous studies have highlighted the potential importance of representing wave processes for providing improved ocean model simulations. For example, Brown et al. (2011) presented case study evidence of tide-surge-wave interactions from a two-way coupled ocean-wave system based on the POLCOMS and WAM models run at a range of horizontal

resolutions at 12 km, 1.8 km and 180 m, focussed on the shallow macrotidal Liverpool Bay region along the north-west England coast during an extreme storm event. Bolaños et al. (2014) extended this work to consider wave-current interactions within the very near-coastal zone in the adjacent Dee Estuary. Reza Hashemi et al. (2015) applied the ROMS-SWAN ocean-wave model coupling in the COAWST (Warner et al., 2010) modelling system on a domain across the north-west European shelf of similar extent to that covered by UKC3. Their assessment focussed largely on the impact of ocean tides on the wave model

performance, with improvements of up to 25% in places where wave-current interaction is significant, wich is of similar magnitude to that demonstrated by Lewis et al. (2018) from the one-way ocean-wave coupling in UKC2. It is worth noting that both Brown et al. (2011) and Reza Hashemi et al. (2015) commented on the disproportionate increase in computational cost incurred through introduction of the coupled system, relative to production of ocean-only or wave-only results. A summary of UKC3 configuration computational costs are provided in Sect. 4.

Staneva et al. (2016a), Staneva et al. (2016b) and Staneva et al. (2017) describe the application of coupling between wave and ocean models and its impact on improving model performance for the German Bight region of the southern North Sea for several extreme events. A change of 20 – 30 cm in the forecast surge level was computed in one case when accounting for



wave forcing in an ocean model, while wave forcing was found to improve the representation of the vertical ocean profile relative to observations.

The following sections describe wave-to-ocean parameterisations applied in the UKC3 coupled configuration. NEMO parameter settings are highlighted where relevant. A list of all symbols used is provided in Appendix A for reference, and

vector quantities are shown in bold.

Appendix B provides a summary of the technical aspects relating to the implementation of wave-to-ocean coupling in the NEMO ocean model and relevant namelist and parameter options. All related ocean model code is now available to the community through an update to the NEMO trunk code resulting from this work (e.g. Law Chune and Aouf, 2018). Researchers and model developers can access this from NEMO vn4.0 (http://nemo-ocean.eu).

The branch of the WAVEWATCH III wave model used for UKW3 and UKC3 has also been adapted to support these developments by providing the new coupling functionality, mainly by calculating and/or adding new coupling fields to the coupling communication, and to the diagnostics. The advantage of also adding the new coupling fields to the diagnostics is that the WAVEWATCH III output can be used as input for NEMO working in forcing mode. Careful tests were made to ensure that the models provided the same output when working in forced and coupled mode, if the information passed to the models

was the same. Comparisons with the WAM wave model code (Komen et al., 1994) have also been made, in particular with relation to the definition of terms in the new wave-modified stress calculations.

**3.1 Momentum modified by drag coefficient (atmospheric forcing modes only)**

The wind stress from the atmosphere at the ocean surface that is transmitted to the ocean is modified due to the wave roughness. In a fully coupled system in which the atmospheric boundary layer is modified by the wave state (e.g. UKC3aow), it is

considered that this effect is simulated through the wave-atmosphere coupling, and so the ocean component is driven by a wave-modified atmosphere. However, for partially coupled (e.g. UKC3ow) or wave forced (e.g. UKO3gw) ocean configurations, it is possible to account for the wave-modified drag in computing the wind stress acting on the ocean.

In the UKO3 shelf sea configuration (with *ln_shelf_flx=.true.* in namelist *namsbc_flx*) the wind components are read, and the wind stress is typically calculated from them using a drag coefficient which is a function of the wind velocity (*nn_drag=0*),

according to Eq. (1) (Smith and Banke, 1975). In the new coupled or wave-forced implementation (*ln_cdgw=.true.*), the shelf sea configuration is also used and the drag coefficient $C_D$ is calculated by the wave model (*nn_drag=1*), and applied as shown in Eq. (2). This formulation can also be used (*nn_drag=2*) with a constant value for $C_D$, set at 0.0015.

$$\tau = \frac{1}{\rho_{\text{ref}}}(0.63 + 0.066|U|)\rho_{air}|U|U \,, \tag{1}$$

$$\tau = C_D\rho_{air}|U|U \,, \tag{2}$$

Alternative implementations of this effect are also available for use with bulk forcing mode configurations of NEMO.

As highlighted by Eq. (1), $C_D$ is a function of wind speed. Figure 2 illustrates the distribution of $C_D$ computed by the UKW3g WAVEWATCH III wave model configuration for a summer and winter month, and Figure 2(h) shows its simulated variation



as a function of forcing wind speed at a selected point in the North Sea. This shows values of between order 0.005 and 0.0025. The *nn_drag=2* default constant of 0.0015 appears to correspond to forcing winds of order 10 ms$^{-1}$. There is a general tendency for the wave simulated $C_D$ to exceed the default *nn_drag=0* formulation (i.e. Eq. 1; Smith and Banke, 1975) for wind speeds in excess of order 5 ms$^{-1}$, and produce lower values for slower wind speeds. Results from UKC3ow and UKW3go simulations

(not shown) also confirm generally low sensitivity of $C_D$ values to the presence of ocean forcing or coupling in the wave model.

**3.2 Momentum fraction transferred to the ocean through wave breaking**

Part of the momentum that the ocean receives from the atmosphere (after taking into account the effect of wave roughness either through wave-atmosphere coupling or modifying the drag coefficient through Sect. 3.1) is stored in surface waves through wave growth or released from the surface waves on wave breaking. The momentum that actually will force the ocean

is therefore a fraction of the atmospheric momentum, which is calculated within the wave model according to Eq. (3). The fraction of atmospheric momentum transferred to the ocean is approximated by the normalised momentum flux variable tauoc (Breivik et al., 2015).

$$\tau_{ocn} = \tau_{atm} - \tau_{wav} + \tau_{wav:ocn} \approx \tau_{atm} * \text{tauoc} , \qquad (3)$$

The following definitions are used to describe each component of the surface momentum budget:

$\tau_{atm}$        - stress applied by atmosphere on ocean surface
        $\tau_{wav}$        - momentum flux absorbed by wave field
        $\tau_{wav:ocn}$    - momentum stored by waves released to ocean through wave breaking
        $\tau_{ocn}$        - water-side stress transmitted into the ocean

Figure 2(a) and Figure 2(d) show the mean simulated tauoc for a summer and winter month, and highlights values tend to lie

in the range 0.95 to 1.05 (i.e. order 5% modification to the atmosphere surface stress due to waves). Largest enhancement can be found along west-facing coastlines, and largest reductions in the lee of land such as downstream of the Scottish islands, in the Irish Sea and along the English Channel. The spatial distribution is broadly consistent between summer and winter months, but with the magnitude of wave modification clearly increased in winter.

**3.3 Stokes-Coriolis drift**

The Stokes drift, caused by finite amplitude waves, creates a relative motion along the wave direction which quickly decays with depth. The NEMO momentum equation is modified to account for the Stokes drift velocity $\boldsymbol{v}_s$, taking into account the Coriolis forcing, as in Eq. (4).

$$\frac{D\boldsymbol{u}}{Dt} = -\frac{1}{\rho_w}\nabla p + (\boldsymbol{u} + \boldsymbol{v}_s) \times f\hat{\boldsymbol{z}} + \frac{1}{\rho_w}\frac{d\boldsymbol{\tau}}{dz}, \qquad (4)$$

As only the surface Stokes drift, $\boldsymbol{v_0}$, is known from the wave model, different parameterizations are used to estimate the change

in the Stokes drift velocity with depth, $\boldsymbol{v}_s(z)$, as a function of the mean wave period, $t_{01}$, significant wave height $H_s$, and peak wave frequency $\omega_p$. Options are controlled by the *nn_sdrift* NEMO namelist parameter.



For *nn_sdrift=0*, the Breivik 2015 parameterization is used (Breivik et al., 2015; (Eq. 120)), with the Stokes drift velocity profile $\boldsymbol{v}_s(z)$ given by Eq. (5). If *nn_sdrift=1*, the Phillips parameterization (Breivik et al., 2016 (Eq. 100)) is applied using an inverse depth scale, according to Eq. (6). An extension can be applied if *nn_sdrift=2* using the peak wave number as calculated by the wave model rather than the inverse depth scale, as shown in Eq. (7).

0: $\qquad \boldsymbol{v}_s(z) = \boldsymbol{v_0}\dfrac{e^{2k_e z}}{1-8k_e z}$ $\qquad\qquad\qquad\qquad k_e = \dfrac{|\boldsymbol{v_0}|}{5.97}\dfrac{16}{2\pi}\dfrac{t_{01}}{H_s^2}$ $\qquad$ (Breivik 2015) , $\qquad\qquad$ (5)

   1: $\qquad \boldsymbol{v}_s(z) = \boldsymbol{v_0}\left[e^{2k_e z} - \beta\sqrt{-2k_e \pi z}\ \mathrm{erfc}\left(\sqrt{2k_e z}\right)\right]$ $\quad k_e = \dfrac{|\boldsymbol{v_0}|}{5.97}\dfrac{16}{2\pi}\dfrac{t_{01}}{H_s^2}$ $\qquad$ (Phillips 2015) , $\qquad$ (6)

   2: $\qquad \boldsymbol{v}_s(z) = \boldsymbol{v_0}\left[e^{2k_p z} - \beta\sqrt{-2k_p \pi z}\ \mathrm{erfc}\left(\sqrt{2k_p z}\right)\right]$ $\qquad k_p = \dfrac{\omega_p^2}{g}$ $\qquad$ (Phillips 2015), $\qquad$ (7)

Figure 2 also shows a summer and winter spatial distribution of the surface Stokes drift velocity and its variability with wind speed.

**3.4 Wave-modified surface roughness**

The ocean surface roughness, which has an effect in the vertical mixing by defining the surface turbulent mixing length scale, can be calculated in different ways. In the Generic Length Scale (GLS) turbulent closure scheme, used in the UKO3 configurations, this is dependent on the choice of parameter *nn_z0_met*. For example, Eq. (8) and Eq. (9) show the simplest approach defining either a constant roughness or constant Charnock parameter via namelist settings.

nn_z0_met = 0:    $z_0 = $ rn_hsro $= 0.02$ m $\qquad\qquad$ [constant roughness] $\qquad\qquad\qquad$ , $\qquad\qquad$ (8)

   nn_z0_met = 1:    $z_0 = MAX\left[\dfrac{\alpha}{g}u_*^2,\ \text{rn\_hsro}\right]$ $\qquad$ [constant Charnock parameter α]    , $\qquad\qquad$ (9)

Rascle et al. (2008) discuss how the roughness length is more physically related to the scale of breaking waves and related eddies responsible for high mixing levels close to the surface, and that it can be related to the significant wave height $H_s$ (or more strictly the windsea wave height).

By default in UKO3 ocean-only configurations, the Rascle et al. (2008) parameterisation for $H_s$ (*nn_z0_met=2*) is used. This estimates the wave age $C_p/u_*$ and subsequently significant wave height $H_s$ as a function of wind speed, through Eq. (10a) and Eq. (10b).

   nn_z0_met = 2:    $\dfrac{C_p}{u_*} = 30\tanh\left(\dfrac{2u_{*ref}}{u_*}\right)$ $\qquad$ [wave age; Rascle et al., 2008] $\qquad\qquad$ (10a)

$\qquad\qquad\qquad z_0 = $ rn_frac_hs $*\left(\dfrac{665}{0.85}\left(\dfrac{C_p}{u_*}\right)^{\frac{3}{2}}\dfrac{u_*}{g}\right)$ [$H_s$ dependent; Rascle et al., 2008] $\qquad\qquad$ (10b)

Here $u_{*ref}$ is a typical friction velocity (0.3 m s$^{-1}$). The surface roughness length $z_0$ is then taken as a fraction *rn_frac_hs* (default value 1.3) of the estimated $H_s$. The appropriate value for this factor for the North-West Shelf region should be reviewed further before consideration of this scheme for operational applications, while further development might consider the coupling of the windsea wave height computed by WAVEWATCH III explicitly.



When using wave forcing or coupling (*nn_z0_met*=3), the same approach is used but with the wave model significant wave height replacing the Rascle et al. (2008) estimate, according to Eq. (11).

$$\text{nn\_z0\_met} = 3: \quad z_0 = MAX[\text{rn\_frac\_hs} * H_s(x,t), \quad \text{rn\_hsro}] \qquad [\text{wave model } H_s(x,t)], \tag{11}$$

## 4 Performance of UKC3 and the impact of coupling

### 4.1 Evaluation framework

Lewis et al. (2018) described the development of an evaluation framework to understand the performance of model components run within coupled systems relative to uncoupled approaches. All coupled and uncoupled configurations defined in Table 2 are provided as rose suites (http://metomi.github.io/rose/doc/rose.html) and version controlled under the Flexible Configuration Management (FCM) system (http://metomi.github.io/fcm/doc/). A number of different options for initial conditions or forcing are available for each configuration (Table 2), which are enabled within a given suite by setting the relevant RUNID environment variable.

### 4.2 Model experiments

The focus of UKC3 system evaluation discussed in this paper is on a series of four coupled and forced simulation experiments, each of approximately 1-month duration. This enables assessment across a variety of meteorological conditions within a given month and of ocean and wave results across a spring-neap tidal cycle, and covers evaluation at different times of the year. In order to capture a range of conditions, experiments are conducted for the following periods:

    a)   'Spring': 30 March 2014 – 19 April 2014

    b)   'Summer': 30 June 2014 – 30 July 2014

    c)   'Autumn': 30 September 2014 – 30 October 2014

    d)   'Winter': 30 January 2015 – 28 February 2015

This approach complements the analysis of Lewis et al. (2018) who considered a number of relatively short 5-day case study simulations across a range of conditions. By extending the simulation period, it is hoped to provide a more comprehensive evaluation of system performance and to establish whether any long term drifts develop in any component or have an impact on the coupled system overall. These experiments therefore represent the first time for the UK regional coupled prediction system to be run for such duration, and provides a useful check on the feasibility of applying the UKC3 or its successor systems for longer term applications such as generation of climate scenarios. In general, it is found that the coupled system remains stable over the month-long simulation period across ocean, wave and atmosphere components, with no serious model drifts found, even without data assimilation. This gives confidence in the scientific validity of the configurations developed and suggests these to be suitable tools for conducting even longer duration research runs.



Table 10 lists the model simulations conducted for each period. Given the increased computational cost of running model experiments over longer periods, only a subset of the possible system configurations defined in Table 2 are considered here. Coupled model results from fully (UKC3aow) and partially (UKC3ao, UKC3owg) coupled mode simulations are compared with the UKA3u, UKO3g and UKW3g control simulations, as these are considered to be the most analogous to typical

operational configurations currently in use.

It is considered most efficient to focus results on a relative evaluation between coupled and uncoupled simulations of the UKC3 system over the selected periods, rather than on a comparison between the UKC2 and UKC3 releases. This approach helps to isolate the impact of the new coupling capabilities within UKC3 from any other code or configuration updates between versions, and provides a more relevant summary of the relative performance of the coupled system relative to current

operational approaches usin more recent science configurations and code.

All simulations are initialised with the same initial conditions relevant to each model component, and the lateral boundary conditions applied are common across simulations, irrespective of the mode of running.

The following analysis compares model outputs to a variety of *in-situ* observations taken from the Met Office operational archive. Figure 1 illustrates the typical data availability, as available for assessment of one day of the 'Summer' 2014

experiment. Further details on the observing networks are provided by Lewis et al. (2018).

### 4.3 Ocean component results

Figure 3 shows a summary of experiment-mean ocean model SST results across each simulation period, comparing fully coupled UKC3aow, partially coupled UKC3ao and forced-mode UKO3g configurations. The sensitivity of SST to coupling is highest during summer months as expected (e.g. Lewis et al., 2018).

The mean differences UKC3aow – UKO3g represent the impact of full atmosphere-ocean-wave coupling relative to a free-running ocean-only configuration. To first order, differences are therefore a combination of the impact of wave forcing and feedbacks on the ocean, the impact of a change both in meteorological forcing resulting from increased atmospheric resolution from global (~17 km) to regional (1.5 km) scale and the effect of three-way coupled feedbacks between ocean and atmosphere, ocean and waves. In April, July and October runs, the impact of full coupling is a mean reduction of SST by typically 0.2 K,

but by up to 1 K during July 2014 (Fig. 4(d)). The relative impact of wave feedbacks on both the ocean and atmosphere is illustrated in Figs. 4(b),4(e),4(h) and 4(k) by comparing UKC3aow with UKC3ao. This highlights a general tendency for wave coupling to cool the simulated SST, by up to 0.5 K, which is found to be mostly driven by a relatively reduced surface drag in April, July and October at least (e.g. Figure 2). This effect can be replicated in the partially coupled UKC3owg configuration through the wave-modified surface drag (Sect. 3.1).

The comparison between model results and *in situ* SST observations presented in Fig. 3 shows a general improvement in RMSE statistics, particularly at the near-coastal buoys but also more widely, for UKC3aow relative to UKO3g. Time series of the average ocean model bias for SST through each simulation are shown in Fig. 4. This highlights that the UKO3g ocean only



simulation is generally biased warm for each season, and by up to 1 K during the July 2014 run. Note that all simulations are initialised from a multi-annual run of the UKO3g ocean configuration. For the July 2014 experiment, while UKO3g results maintain the initial bias of order 1 K too warm throughout the month-long simulation, the initial bias is eroded over the first week or so of simulations to be order 0.2 K too warm in both UKC3ao and UKC3aow runs, and is further reduced in the last

week of the simulation. Similar features but of smaller magnitude can be seen during April 2014 and October 2014 experiments. This result is found to be a function of both the spatial and/or temporal resolution change (noting that the UKO3g forcing is obtained from operational archives with data assimilation) in the atmosphere forcing, and a result of an improved atmospheric state due to the dynamic coupling to the ocean (and wave) component. The magnitude of improvements due to coupled relative to observations is further highlighted for each month in Fig. 5(b), 5(d) and 5(f) which show the time series of

the absolute model bias relative to the UKO3g control.

The contribution of wave processes to the reduction in SST bias can be determined in fully coupled mode by comparing UKC3aow and UKC3ao results (dark red relative to light red in Fig. 5), and in partially coupled mode with consistent atmospheric forcing by comparing UKC3owg with ocean only UKO3g (blue relative to grey line in Fig. 5). This shows representation of wave processes to improve the agreement of simulated SST with observations, but also that this contribution

is estimated to be of order 10% of the differences between fully coupled and ocean only. A larger relative improvement is also found in forced mode (UKC3owg – UKO3g) than found between UKC3aow and UKC3ao in April and July 2014 at least.

Results for a winter month (February 2015) show the impact of full coupling and wave feedbacks to be more isolated to near-coastal areas, and while improvements in RMSE for UKC3aow relative to UKO3g can be seen in Fig. 4(l), the relative difference in mean bias in Fig. 5(h) fluctuates through the month (but typically within 0.1 K). In this case, the impact of wave

feedbacks is generally very similar between UKC3aow and UKC3owg simulations relative to UKO3g.

This discussion highlights the potential of regional coupled systems to deliver improved simulation of the ocean state, and this development is being tested for implementation within the framework of the EU Copernicus Marine Environment Monitoring Service (CMEMS) for the North West European Shelf region for example.

## 4.4 Wave component results

It is widely known that the quality of the wave model results is critically dependent on the quality of the wind forcing, or that applied via atmosphere-wave coupling (e.g. Cavaleri et al., 2018). The benefit of coupling on wave results was therefore characterised by Lewis et al. (2018) for case study simulations through comparing UKC2aow results with the UKW2h control using a comparably high resolution wind forcing, while it was found to be generally difficult to improve on the performance of the wave-only simulations forced by operational archive global resolution MetUM winds in the UKW2g configuration. The

same characteristics are found for UKC3 results in this study, in which the fully coupled UKC3aow runs are compared only with partially coupled UKC3owg and forced mode UKW3g simulations in which the wind forcing is provided by the global resolution operational MetUM archive. Figure 6 illustrates the differences in wind forcing during the October 2014 experiment,



which is representative of results for other months. The impact of atmosphere model resolution can be seen in particular around the coasts, where the UKC3aow winds exceed those from the global MetUM forcing. This is a combination of the effect of increased drag over land impacting a broader region in the global-scale forcing than at high resolution, and from including currents and tidal feedbacks in the lower boundary condition in the coupled configuration.

Wind speeds tend to be slightly reduced in regions away from coastlines across the north-western and south-western approaches to the UK, and across the southern North Sea. These features are generally replicated during other months. Comparison with available in situ wind observations located in the ocean (Fig. 5(d)), noting these are generally located away from near-coastal areas, demonstrate a consistent reduction in quality of winds at high resolution in UKC3aow relative to the global archive MetUM forcing. Figure 6 presents a more quantitative analysis through each experiment, and shows that on

average, the UKW3g (and UKC3owg) forcing is biased fast by up to 1 m s$^{-1}$ during each run across the domain. The fully coupled UKC3aow winds are by contrast biased fast by up to 2 m s$^{-1}$ during the four periods, but with much greater variability in the magnitude and sign of the bias through time than found for the UKW3g MetUM forcing. The fast bias is consistent with the recent analysis by Jiménez and Duidha (2018) who compared WRF model simulations with observations from the FINO towers located in the southern North Sea.

Figure 5(c) demonstrates the close link between wind speed differences between configurations and their impact on significant wave height, Hs, with increased wind speeds in UKC3aow over the northern North Sea on average tending to drive waves of increased magnitude, and reduced wave heights to the west of the UK in the fully coupled UKC3aow simulations relative to the wave-only UKW3g control. Figure 5(e) also highlights an expected general reduction in the level of agreement between UKC3aow wave height simulations with *in situ* observations relative to UKW3g. However, even given the degraded wind

speed forcing, relative improvement can be seen at a number of near-coastal sites along the south-western English Channel coast.

To isolate the impact of wave-ocean feedbacks from the wind forcing, Fig. 5(h) and Fig. 5(k) compares UKC3owg with UKW3g results. This shows a general reduction of significant wave height in most areas, but a region of slightly enhanced wave heights along the northern half of the English Channel, associated with wave-current interactions. The impact of current-

wave interactions in the near coastal zone was also discussed by Lewis et al. (2018a). In contrast to Fig 5(e), wave-ocean interaction is shown to have a clear beneficial impact on the agreement between observed and simulated wave height in Fig 5(k).

The time series of average Model – Observation bias in Hs shown in Fig. 7 during each experiment period reflect a tendency for the global-scale wind driven UKW3g and UKC3owg simulations to under-predict significant wave heights across all

months considered by up to 0.2 m on average. The sensitivity to coupling is found to be generally consistent across the different months. The increase in wave heights through enhanced winds in the fully coupled UKC3aow system tends to improve the bias for some periods during each month. However, there are as many periods when UKC3aow results become biased high relative to observations. On average, the impact of representing ocean-wave feedback processes in UKC3owg is shown to be



relatively small (within 0.05 m) in comparison with the wind-related UKC3aow differences, with no clear improvement or degradation in performance across the experiments considered.

Figure 5 also shows the impact of coupling on wave mean period during the October 2014 experiment. Results are particularly improved along the south England coast, for both UKC3aow (Fig. 5(f)) and UKC3owg (Fig. 5(l)). This is consistent with the
case study results of Lewis et al. (2018a), and with Palmer and Saulter (2016). They found that the inclusion of surface currents in the Met Office UK4 operational wave system improved the representation of swell in this region. This was largely due to the refraction of long period waves towards the coast dur to wave current interaction (evident in Fig. 5(c) and Fig. 5(i)). The reduction in quality of mean wave period results against observations along the eastern England coast in Fig. 5(f) is not apparent for UKC3owg (Fig. 5(l)), suggesting that the wind speed errors continue to dominate here.

Figure 8 shows the time series of model bias of mean wave period through each experiment. This highlights all model results producing waves that are longer period than observed at near-coastal sites for much of the time. Both the fully coupled UKC3aow and partially coupled UKC3owg simulations provide reduced biases.

### 4.5 Atmosphere component results

Comparing the UKC3aow and UKC3ao atmosphere results with the UKA3u atmosphere-only control simulation provides a
strong test of the system. Whereas the coupled system SST evolves according to the free running NEMO ocean model component, the surface forcing in the UKA3u control simulation has a daily updating analysis-based (OSTIA; Donlon et al., 2008) SST in closer agreement with observations on average. The key limitation in UKA3u is therefore that the system has no information on the diurnal cycle of SST.

Figure 9 summarises differences in the surface temperature across both land and sea in UKC3aow fully coupled and UKC3ao
partially coupled simulations, relative to UKA3u. The distribution of monthly mean differences are quite varied between each month considered, reflecting both the seasonal variability in the quality of the coupled system ocean initial condition and its subsequent evolution (see Sect. 4.3). In April 2014, October 2014 and February 2015 experiments, the coupled model SST tends to be warmer than OSTIA, while in July 2014 SST tend to be cooler away from the permanently mixed southern North Sea region and in the Celtic Sea. The impact of wave coupling processes on the SST evolution in UKC3aow relative to UKC3ao
is as shown in Fig. 4 and discussed in Sect. 4.3. The comparison of results with in situ observations in Fig. 9 shows that in general the persisted SST from OSTIA better matches observations across much of the domain, as might be anticipated. However, nobable areas where the UKC3aow surface temperatures are improved relative to OSTIA can be seen along at least some coastal regions in each month considered, where it is known that the satellite-based analysis product used in UKA3u is likely to be degraded by proximity to the coastline.

Noting the relatively large number of near-coastal sites contributing to the assessment, the timeseries of average bias during each month in Fig. 10 also show reasonably close agreement (typically within 0.5 K; slightly lower for the fully coupled UKC3aow case) of the coupled SST results with *in situ* observations, and extended periods of time when results show improved





SST in the coupled simulations relative to UKA3u. The results for July 2014 in particular also highlight a diurnal cycle of SST bias in UKA3u, reflecting the daily persisted SST. In contrast, the representative dynamical representation of the diurnal SST cycle in the NEMO ocean model component results in a relatively smooth variation of the average bias for UKC3aow and UKC3ao.

The key question to address is on the extent to which these differences in the surface temperature forcing, along with resulting differences in the surface momentum budget, change the coupled system meteorology relative to UKA3u. The spatial distribution of monthly mean differences in air temperature at 1.5 m above the surface due to coupling shown in Fig. 11 closely reflect the distribution of differences in mean surface temperature (Fig. 9). Differences over the ocean are dominated by the bias between the coupled ocean simulation and OSTIA, with the contribution due to diurnal cycle differences masked in the

monthly mean measures presented. Differences over land are thought to result from a combination of advection of relatively warmer/cooler air from over a nearby warmer/cooler ocean between simulations, and of resulting differences to boundary layer and cloud development (e.g. Fallmann et al., 2017). Figure 11 also shows a general degradation of the agreement with in situ observations of 1.5 m air temperature over much of the domain, consistent with the SST results. However, specific regions can be seen where the fully coupled UKC3aow results have reduced RMSE in each month. The time series of average bias in air

temperature across all observation sites shown in Fig. 12 is dominated by errors over land, which demonstrates a clear diurnal signal of the bias in all simulations (e.g. Bush et al., 2018). Simulated temperatures are relatively too cool during daytime and too warm at night. The impact of coupling during the periods considered in this study is, in general, to consistently shift the bias (typically warmed) at all times of the day, such that the bias is apparently 'improved' relative to UKA3u during daytime but degraded at nighttime when UKA3u has a warm bias. Longer periods of improved air temperature results from UKC3aow

are apparent during April, October and February experimnets.

    The distribution of experiment-mean 10 m wind speed changes due to model coupling are presented in Fig. 13. The differences between UKC3aow and UKC3ao highlight that the main impact of wave coupling across all months is a reduction in the monthly mean wind speed, by up to 0.5 m s$^{-1}$. This is consistent with the wave model computed Charnock parameter tending to exceed the default assumed value of 0.011 across much of the domain (e.g. Fig. 2). This effect is most prominent during

stormier periods, such as the October 2014 experiment.

    Figure 13 shows a more varied spatial distribution of differences between UKC3aow and UKA3u, due to more substantial difference in SST between the simulations, in addition to the Charnock parameter coupling. Comparing Fig. 13 with Fig. 9 suggests that to first order, areas of relatively increased winds align with regions of relatively enhanced sea surface (and air) temperatures, and those with reduced wind speeds align with regions of reduced temperatures – i.e. the mean SST and wind

speed anomalies are positively correlated. A growing literature has developed over recent years on the extent to which ocean temperature deviations drive atmospheric responses or vice versa at mesoscales in different regions of the world (e.g. Small et al., 2008; Gemmrich and Monahan, 2018). Figure 14 provides an initial assessment of the variability of monthly mean wind speed differences between simulations over the sea with differences in the near-surface temperature gradient (estimated as $T_{air}$



– SST) and with differences in surface currents. This shows that where mean (land and sea) surface temperatures are increases (decreased) in UKC3aow relative to UKA3u, the impact tends to be a reduction (increase) in the near surface temperature gradient. According to surface layer theory, under increasingly unstable conditions (change in stability < 0), the surface drag is increased and the near surface wind speed is expected to increase (change in wind speed > 0). This mechanism is at least

partly demonstrated by the variability of mean wind speed and near-surface temperature gradient differences shown in Fig. 14(b). This contrasts with no clear relationship evident between wind speed and surface current differences in Fig. 14(c). Corresponding results for UKC3aow and UKC3ao differences in Fig. 14 also show no strong dependencies between variables, highlighting the wind speed differences to be largely driven by the use of the wave model Charnock parameter in UKC3aow. In common with air temperature results in Fig. 12, the time series of wind speed model bias for all simulations shown in Fig.

15 also demonstrate a diurnal cycle across all configurations, with winds too strong by up to 1 m s$^{-1}$ on average during daytime and slightly too weak at night. As an average across all sites, the relative impact of coupling is less pronounced than found for temperature variables, with changes due to ocean-atmosphere coupling within 0.1 m s$^{-1}$ for much of the periods considered. The impact of Charnock coupling is evident with improved results for UKC3aow relative to UKC3ao, and extended periods where the UKC3aow winds are improved relative to UKA3u, despite the improved SST specification in UKA3u. Results are

consistently improved for UKC3aow during the February experiment for example. In contrast, from 15 October 2014, Fig. 15 shows both UKC3ao and UKC3aow simulations are degraded relative to the UKA3u control, coinciding with a period of relatively poorer surface temperature bias.

Despite the strong test set by comparing UKC3aow and UKC3ao performance to the atmosphere-only UKA3u control, these results demonstrate that representative simulations of the atmosphere can be performed in fully coupled mode at convective

scales. It is clear that inclusion of the two-way feedbacks between surface waves and both ocean and atmosphere components provides some benefit over only including ocean-atmosphere feedbacks. The impact of coupling on the atmospheric boundary layer and associated features such as cloud development and near surface visibility are the subject of ongoing research, focussing on more specific case studies periods and regions of interest (e.g. Fallmann et al., 2018).

### 4.6 Computational resource

Table 11 summarises the typical computational resource usage and run times for a day of simulation on the Met Office Cray XC40 for each configuration used. No system optimisation has been performed for UKC3, relative to the UKC2 configuration, and further opportunities for system optimisation remain that will be pursued as part of the future UKC4 development.

### 5 Discussion and ongoing development

This paper has provided an update on the evolution of a regional atmosphere-ocean-wave coupled prediction system for the

UK at km-scale resolution. The UKC3 system represents a further important development through the successful introduction





and testing of a number of wave-to-ocean feedback processes and related data exchanges, such that for the first time UKC3aow provides a truly coupled system with two-way feedbacks represented between all components.

The four monthly experiments presented here also represent the first runs conducted of the UKC3 (or UKC2) system of extended duration beyond the 5-day case study duration simulations described by Lewis et al. (2018), or used in the studies of

Fallmann et al., (2017) and Fallmann et al., (2018). That the results continue to show robust and representative predictions across atmosphere, ocean and surface wave components in coupled mode throughout these periods provides confidence in the scientific integrity of these tools, and of their suitability for application over longer-timescales in future. The quality and limitations of the UKC3 system relative to uncoupled approaches has been discussed.

A number of summary results have been presented in this paper, either as monthly mean differences between coupled and

uncoupled simulations or time series of the average biases between model and observations across a number of *in situ* observing sites through each experiment. This only provides an initial and top-level snapshot of model performance. Evidence from these experiments and Lewis et al. (2018) indicate that representing feedbacks between components adds skill to the modelling system in specific situations and at certain locations, such that any impacts can be damped in a presentation of results aggregated in time or space. In order to guide development priorities and improvement, further analysis is required to examine

the specific locations and time periods for which model performance is particularly degraded or improved by coupling across any of the components of interest. This case study mode work is ongoing and will be reported through further publication.

The biggest impact of simulating in coupled mode, relative to the uncoupled configuration most analogous to current operational approaches, has been demonstrated in the ocean component SST, with a marked decrease in model bias achieved through April, July and October 2014 experiments, in part due to inclusion of wave processes in the ocean, and largely as a

result of using the coupled high-resolution atmospheric forcing rather than the operational global-scale MetUM forcing.

Such general improvements were not apparent for the wave model results in contrast, given the comparison to a wave-only model forced by winds from the same operational global-scale MetUM forecasts. The improved quality and lower variability of wind forcing into UKW3g continues to drive improved quality simulation of wave height and mean period at most sites. For a given atmospheric forcing, the positive impact of representing wave-ocean feedbacks on results has been demonstrated

however. Better understanding and improving the impact of atmosphere spatial (and temporal) resolution of wind forcing or coupled information remains a priority for future development, in order that wave model results within the fully coupled system might be improved relative to wave-only approaches forced by global-scale winds.

The atmosphere component results also highlight that the coupled system inherits a number of the underpinning model biases associated with convective-scale MetUM configurations, with representation of a diurnal temperature cycle through ocean-

atmosphere coupling and dynamic evolution of surface roughness through wave-atmosphere coupling neither substantially degrading or improving predictions, on average. More detailed tesing of the boundary layer sensitivity to surface processes at convective scale will be of benefit for future improvement.



Having developed a fully coupled UKC3 research tool, and demonstrated its application over an extended period of time, the priorities for ongoing research and improvement can be summarized as follows.

**Improving predictability**

Improving skill beyond the UKC3 capability presented in this paper will require two key developments. Firstly, a number of

the parameterization and parameter assumptions embedded across the MetUM atmosphere, NEMO ocean and WAVEWATCH III wave model codes and their configuration in the regional system will need to be re-examined and challenged. The UKC3 configuration still represents a 'first look' implementation, in which a number of the underpinning assumptions and parameter choices established with uncoupled mode forcing or boundary data continue to exist. For example, consistent treatment of the atmospheric surface layer momentum budget across all codes has been an area of focus through the implementation of the

wave-modified drag feedbacks within UKC3. The extent to which coupled mode running of these components are working against existing tuning is to be assessed. The influence of wind forcing on the skill of the wave model is a key example. The coupling exchange frequency is a further area deserving further study and optimization, given that the hourly coupling used in the UKC3 experiments presented in this paper is long relative to most studies discussed in Sect. 1.

Secondly, it is appropriate to begin developing the regional coupled system in an assimilative context, in order to improve the

initial condition errors inherent within the current experimental design, as seen in the ocean SST results here for example. This is also a key step towards consideration of such systems for operational applications. Wada and Kunii (2017) have recently demonstrated the successful application of a regional mesoscale strongly coupled atmosphere-ocean data assimilation, implementing a local ensemble transform Kalman filter, for a tropical cyclone case study. For the UK regional coupled system development, it is more likely that a weakly coupled assimilation approach will be followed building on experience of

developing this system for global coupled NWP applications at the Met Office (e.g. Lea et al., 2015). It is not immediately clear how moving to an assimilative framework will modify any sensitivity to coupling, if at all. One hypothesis is that the observational constraint will reduce the relative impact of representing feedback processes in the system. Conversely, improving the background state through improving the physics representation may in fact enhance the impact of data assimilation within the system by reducing the background errors. Research is therefore required in this area. One particularly

beneficial development that will support this activity in future is the implementation of analogous UKV atmosphere (e.g. Bush et al., 2018) and AMM15 ocean configurations (e.g. Graham et al., 2018) for operational forecasting at the Met Office using common physics settings, domain extent and grid definition to those used in UKC3. These provide the potential for operational analyses and boundary conditions for application in future regional coupled research activities.

**Towards integrated environmental prediction capability**

The focus of this paper has been on the physical coupling processes across atmosphere, ocean and wave components, with the driver of improving predictability through improved physical process representation. The vision for regional coupled




prediction systems is that they can also provide a framework in which to represent interactions and feedbacks between physical, hydrological and biogeochemical cylces and processes at km-scale. The initial focus in this context remains to demonstrate coupled prediction of the water cycle across atmosphere, land and ocean. Improvements to the convective rainfall representation and improved accuracy of quantitative precipitation forecasts in the UKC3 (RA1 physics) configuration relative

to that in the UKC2 system (Bush et al., 2018) forms a key foundation of turning that potential into a reality. Further improvements to the hydrological capability in the JULES land surface model are also planned (e.g. Martinez et al., 2018), and their impact will be documented in the context of developing the UKC4 system.

Feedbacks between physical and biogeochemical processes in the ocean will also be introduced. A first required step is the technical implementation of the ERSEM (European Regional Seas Ecosystem Model; Butenschön et al., 2016) marine

biogeochemical model as a new coupled component into the UKC4 system.

**Longer term considerations**

The potential for delivering consistent natural hazard warnings across the scope of atmosphere, land, ocean and wave components was introduced in Sect. 1. Application of the UKC3 and its subsequent iterations to demonstrating this concept is still required. A key part of realizing this vision in a more operational context, particularly for hydrological and surge hazards

for example, will be the requirement to develop an ensemble of regional coupled predictions. Consideration will need to be given on how to generate a regional ensemble with adequate spread in ocean and land surface states in the short term, with opportunities again to build on experience from the development of global-scale coupled NWP (e.g. Tennant and Beare, 2014). Incremental improvements to the capability and application of the UK regional coupled system will therefore continue over coming years, with the UKC3 system configuration providing an important milestone along that research journey. This effort

will continue to require a multi-disciplinary approach, working in open collaboration both in the UK and with other groups around the world.





## Code availability

### *Intellectual property*

Due to intellectual property right restrictions, neither the source code or documentation papers for the Met Office Unified Model or JULES can be provided directly. All model codes used within the UKC3 configuration are accessible to registered researchers, and links to the relevant code licences and registration pages are provided for each modelling system below. All code used can be made available to the Editor for review. Supplementary material to this paper does include a set of Fortran namelists that define the atmosphere, land, ocean and wave configurations in UKC3 simulations.

### *Obtaining the Met Office Unified Model*

The Met Office Unified Model (MetUM) is available for use under licence. A number of research organisations and national meteorological services use the MetUM in collaboration with the Met Office to undertake basic atmospheric process research, produce forecasts, develop the MetUM code and build and evaluate Earth system models. For further information on how to apply for a licence see http://www.metoffice.gov.uk/research/collaboration/um-partnership. The MetUM vn10.6 trunk code and associated modifications for UKC3 are available to registered researchers via a shared MetUM code repository, which can be accessed via https://code.metoffice.gov.uk/trac/um/wiki. Details of the separate code branches with modifications for UKA3 and UKC3 are documented in the Supplementary Material. A copy of the merged MetUM code used for UKC3 is provided at https://code.metoffice.gov.uk/trac/utils/browser/ukeputils/trunk/gmd-2018/uka3/um to support collaboration.

### *Obtaining JULES*

JULES is available under licence free of charge. For further information on how to gain permission to use JULES for research purposes see http://jules.jchmr.org. The JULES vn4.7 trunk code and associated modifications for UKC3 are then freely available on the JULES code repository, which can be accessed via https://code.metoffice.gov.uk/trac/jules/wiki. Details of the separate code branches with modifications for UKA3/UKL3 and UKC3 are documented in the Supplementary Material. A copy of the merged JULES code used for UKC3 is provided for reference and to support collaboration at https://code.metoffice.gov.uk/trac/utils/browser/ukeputils/trunk/gmd-2018/uka3/jules.

### *Obtaining NEMO*

The model code for NEMO vn3.6 is freely available from the NEMO website (www.nemo-ocean.eu). After registration the FORTRAN code is readily available to researchers. Modifications to the NEMO vn3.6 trunk for UKC3 are also freely available as a copy of the merged code branches at https://code.metoffice.gov.uk/trac/utils/browser/ukeputils/trunk/gmd-2018/uko3/nemo. A list of the NEMO compilation keys applied on building the merged NEMO code is provided in the Supplementary Material. Also provided are details of the separate code branches with modifications for UKO3 and UKC3.

### *Obtaining WAVEWATCH III*

WAVEWATCH III® is distributed under an open source style license to registered users through a password protected distribution site. The licence and link to request model code can be found at the NOAA National Weather Service Environmental Modeling Center webpages at http://polar.ncep.noaa.gov/waves/wavewatch/. The model is subject to



continuous development, with new releases generally becoming available after implementation of a new model version at NCEP. Research model versions may also be made available to those interested in and committed to basic model development, subject to agreement.

The WAVEWATCHIII code base is distributed by NOAA under an open source style licence via
http://polar.ncep.noaa.gov/waves/wavewatch/wavewatch.shtml. Interested readers wishing to access the code are requested to register to obtain a license via http://polar.ncep.noaa.gov/waves/wavewatch/license.shtml. Model codes used in the UKC3 system are maintained under configuration management via a mirror repository hosted at the Met Office, and can be made available to researchers for collaboration on request, given prior approval to access WAVEWATCH III from NOAA. This is provided at https://code.metoffice.gov.uk/trac/utils/browser/ukeputils/trunk/gmd-2018/ukw3. The Supplementary Material
provides a list of the WAVEWATCH III compilation switches applied on building the wave model code.

*Obtaining OASIS3-MCT*

OASIS3-MCT is disemminated to registered users as free software from https://verc.enes.org/oasis.

*Obtaining Rose*

Case study simulations and configuration control namelists were enabled using the rose suite control utilities. Further
information is provided at http://metomi.github.io/rose/doc/rose.html, including documentation and installation instructions.

*Obtaining FCM*

All codes were built using the fcm make extract and build system provided within the Flexible Configuration Management (FCM) tools. Met Office Unified Model and JULES codes and rose suites were also configuration managed using this system. Further information is provided at http://metomi.github.io/fcm/doc/.

**Data availability**

The nature of the 4D data generated in running the various UKC3 experiments at 1.5 km resolution requires a large tape storage facility. These data is of the order tens of Tb. However, the data can be made available upon contacting the authors. Each simulation namelist and input data are also archived under configuration management, and can be made available to researchers to promote collaboration upon contacting the authors.
Ocean bathymetry was obtained from the EMODnet Portal: EMODnet Bathymetry Consortium, EMODnet Digital Bathymetry (DTM), EMODnet  Bathymetry (September 2015 release).




**Appendices**

**Appendix A – List of symbols**

| Symbol | Units | Description | Equation reference |
|---|---|---|---|
| $C_p$ | m s$^{-1}$ | Wave phase speed | 12 |
| $c_D$ | - | Surface exchange coefficient for momentum | 2 |
| $f$ | - | Coriolis parameter | 4 |
| $g$ | m s$^{-2}$ | Acceleration due to gravity | 7,9,10 |
| $H_s$ | m | Significant wave height | 5,6,11 |
| $k_e$ | - | Inverse depth scale for Stokes drift velocity profile | 5,6,7 |
| $t$ | s | Time coordinate | 4 |
| tauoc | - | Normalised ocean to atmosphere stress fraction | 3 |
| $t_{01}$ | s | Wave period | 5,6 |
| $\boldsymbol{U}$ | m s$^{-1}$ | Atmospheric wind speed | 1,2 |
| $\boldsymbol{u}$ | m s$^{-1}$ | Ocean current speed | 1,2 |
| $u_*$ | m s$^{-1}$ | Surface friction velocity | 9,10 |
| $\boldsymbol{v_s}$ | m s$^{-1}$ | Stokes drift velocity | 4,5,6,7 |
| $\boldsymbol{v_0}$ | m s$^{-1}$ | Surface Stokes drift velocity | 5,6,7 |
| $z$ | m | Vertical coordinate | 4,5,6,7 |
| $z_0$ | m | Surface roughness length | 8,9,10,11 |
| $\alpha$ | - | Wave-dependent Charnock coefficient | 9 |
| $\rho_{air}$ | kg m$^{-3}$ | Air density | 1,2 |
| $\rho_{ref}$ | kg m$^{-3}$ | Surface air density | 1 |
| $\rho_w$ | kg m$^{-3}$ | Ocean surface density | 1 |
| $\boldsymbol{\tau}$ | N m$^{-2}$ | Surface stress vector | 1,2 |
| $\tau_{atm}$ | N m$^{-2}$ | Stress applied by atmosphere on ocean surface | 3 |
| $\tau_{ocn}$ | N m$^{-2}$ | Water-side stress transmitted into ocean | 3 |
| $\tau_{wav}$ | N m$^{-2}$ | Momentum flux absorbed by wave field | 3 |
| $\tau_{wav:ocn}$ | N m$^{-2}$ | Momentum released by waves to the ocean | 3 |
| $\omega_p$ | kg m$^{-3}$ | Wave peak frequency | 7 |





## Appendix B – Technical details of NEMO ocean model wave coupling code implemented for UKC3

In parallel with developing the UKC3 coupled configuration, the relevant NEMO ocean model code for wave coupling has been implemented in the NEMO trunk code, in close collaboration as part of the NEMO Wave Working Group, and supported through the Ocean-Wave-Atmosphere Interactions in Regional Seas (OWAIRS) Copernicus Marine Service Evolution project.

This capability is provided at from NEMO vn4.0, including:

- Consolidation of disparate wave science developments from contributing groups into a common code, including support for all those described in Appendix III.

- Support for required wave variables to be passed to the ocean model consistently whether in forced (file passing; core or direct flux forcing) or coupled (OASIS3-MCT library passing) mode,

- Treatment of potentially different land/sea masks across ocean and wave models,

- Removal of implicit assumption in NEMO that, when working in coupled mode, an atmospheric model is always coupled to NEMO,

### B.1 NEMO ocean model wave coupling/forcing namelist switches

To activate wave physics in NEMO in coupled mode it is necessary to specify the same namelist variables as when running
wave physics in forced mode (see below). In addition, it would also be necessary to set the variables *ln_cpl* and/or *ln_wavcpl* to .true. in the namelist *namsbc*, while *ln_mixcpl* should only be .true. if there is mixed forced runs with coupled atmosphere. Specifying *ln_wavcpl=.true.* is also necessary if the coupling is only performed to send fields from the ocean to the wave model. Remember that when running with wave physics it is possible to receive some wave fields via forcing and others via coupling. The list of new NEMO namelist variables is:

Namelist *namsbc*: Switch **ln_wave**: activates wave physics in both forced and coupled mode

Namelist *namsbc_wave*:

**ln_sdw**:

> modifies the surface vertical velocity due to Stokes drift; the necessary forced/coupled fields required for this option could be: wave height, the two components of the surface Stokes drift, the mean wave period, and the peak frequency.
> The specific parameterization for the calculation of the vertical Stokes drift from the surface velocity components is determined by the variable *nn_sdrift* (see below).

**ln_stcor**:

> if *ln_sdw* is .true., it activates the Stokes Coriolis term; no new fields need to be read

**ln_cdgw**:

> reads the neutral drag surface coefficient instead of calculating it; the field needed for this option is the surface drag coefficient. The way the momentum is calculated from the wind components is controlled by the variable *nn_drag*





(see below). If this option is active in direct forcing of coupled mode, the variable *ln_shelf_flx* (namelist namsbc_flx) must be set to .true. in order to read winds instead of momentum from the input.

***ln_tauoc***:

introduces a correction to the ocean stress based in the stress adsorbed by the waves; the necessary field for this option

5        is the fraction of stress that goes into the ocean

***ln_phioc***: [not used in UKC3]

adds the wave breaking mixing effect to the ocean; the necessary field for this option is the wave to ocean energy. The particular wave mixing TKE boundary conditions are controlled with the variable *nn_wmix* (see below)

***ln_rough:***

sets the surface roughness length equals to the significant wave height (making *nn_z0_met=3*); the necessary field for this option is the significant wave height

## B.2 NEMO ocean model wave coupling/forcing namelist parameters

A number of NEMO namelist variables need to be set depending on the values of the *namsbc_wave* switches described in Sect. B.1:

***nn_drag*** (namelist namsbc; relevant when *ln_cdgw=.true.*):

determines how to calculate wind stress from the wind components, in case the wind forcing is received instead of the momentum (variable *ln_shelf_flx* in namelist *namsbc_flx*).

*nn_drag* = 0: wind stress calculated as in the UKMO shelf formulation, with a drag coefficient dependent on wind velocity (see Eq. 2);

*nn_drag* = 1: wind stress calculated with a coefficient that does not depend on the wind velocity, but just on the drag coefficient received via forcing or via coupling (see Eq. 3);

*nn_drag* = 2, wind stress calculated with the same formulation as for *nn_drag*=1, but using a constant, default value of the drag coefficient;

*nn_drag* = 3 and running in core forcing mode, calculates the final drag coefficient using a convergence approach

which needs the total precipitation and specific humidity as input parameters:

***nn_sdrift*** (namelist namsbc_wave; relevant when ln_sdw=.true.):

parameterization to calculate the vertical Stokes drift from the surface components.

*nn_sdrift = 0*, use Breivik 2015 parameterization (Breivik et al., 2015) – Eq. (6)

*nn_sdrift = 1*, use Phillips parameterization (Breivik et al., 2016) – Eq. (7)

*nn_sdrift = 2*, use Phillips parameterization with wave model peak wave number – Eq. (8)

**nn_z0_met** (namelist namzdf_gls; relevant when *ln_rough=.true.*):




method to calculate the surface roughness length. If the compilation keys *key_zdfgls* or *key_esopa* are active and *ln_rough=.true.*, this variable must have a value of 3.

*nn_z0_met* = 0, constant roughness is assumed – Eq. (10)

*nn_z0_met* = 1, constant Charnock formula is assumed – Eq. (11)

*nn_z0_met* = 2, Rascle et al. (2008) parameterisation – Eq. (12)

*nn_z0_met* = 3, Rascle et al. (2008) with simulated wave height – Eq. (13)

**B.3 Forcing or coupled wave fields used by NEMO ocean model**

The list of wave fields that can be received by NEMO in forced (the namelist *namsbc_wave* would have to be completed) or coupled mode (the namelist *namsbc_cpl* would have to be completed) are:

**44. Wave height** (needed if *ln_sdw=.true.* and *nn_sdrift=0,1*) - Namelist variable *sn_swh* in forced mode and *sn_rcv_hsig* in coupled mode - Variable name *O_Hsigwa* in NEMO and *wavehgt* in WWIII

**45. Normalized wave to ocean energy** (needed if *ln_phioc=.true.*) - Namelist variable *sn_phioc* in forced mode and *sn_rcv_phioc* in coupled mode - Variable name *O_PhiOce* in NEMO and *phiwvoce* in WWIII

**46. Stokes drift in the u direction** (needed if *ln_sdw=.true.*) - Namelist variable *sn_usd* in forced mode and

*sn_rcv_sdrfx* in coupled mode - Variable name *O_Sdrfx* in NEMO and *stkdrftx* in WWIII

**47. Stokes drift in the v direction** (needed if *ln_sdw=.true.*) - Namelist variable *sn_vsd* in forced mode and *sn_rcv_sdrfy* in coupled mode - Variable name *O_Sdrfy* in NEMO and *stkdrfty* in WWIII

**48. Mean wave period** (needed if *ln_sdw=.true.* and *nn_sdrift=0,1*) - Namelist variable *sn_wmp* in forced mode and *sn_rcv_wper* in coupled mode - Variable name *O_WPer* in NEMO and *meanwper* in WWIII

**49. Mean wave Number** (needed if *ln_zdfqiao=.true.*) - Namelist variable *sn_wnum* in forced mode and *sn_rcv_wnum* in coupled mode - Variable name *O_WNum* in NEMO and *meanwnum* in WWIII

**50. Stress fraction into the ocean** (needed if *ln_tauoc=.true.*) - Namelist variable *sn_tauoc* in forced mode and *sn_rcv_tauoc* in coupled mode - Variable name *O_TauOce* in NEMO and *taufrac* in WWIII

**51. Surface drag coefficient** (needed if *ln_cdgw=.true.*) - Namelist variable *sn_cdg* in forced mode and

*sn_rcv_wdrag* in coupled mode - Variable name *O_WDrag* in NEMO and *dragcoef* in WWIII

**52. Peak frequency** (needed if *ln_sdw=.true.* and *nn_sdrift=2*) - Namelist variable *sn_wfr* in forced mode and *sn_rcv_wfreq* in coupled mode - Variable name *O_WFreq* in NEMO and *pkfreq* in WWIII

The value of *cldes* (the first parameter of the *namsbc_cpl* variables for wave coupling) can only be 'coupled' or 'none'.

**B.4 NEMO ocean configuration settings required for representing wave processes in forced or coupled modes**

Table B1 lists the changes to NEMO namelist parameters to be set to run a direct forcing run with wave physics enabled using a baseline wave coupling parameterisation with a variable Stokes drift vertical profile. Setting a coupled run is equivalent, but



changing the namelist that read a particular field from forcing for a namelist that couples the same field (for example, changing *sn_cdg* in forced mode by *sn_rcv_wdrag* in coupled mode). To run in uncoupled ocean-only mode, *ln_wave* can be set to .false., and all wave-related NEMO namelist options are ignored.

**Supplement link (will be included by Copernicus)**

5   **Team list**

**Author contribution**

**Competing interests**

The authors declare that they have no conflict of interest

**Disclaimer**

10   **Acknowledgements**

This research has been carried out under national capability funding as part of a directed effort on UK Environmental Prediction, in collaboration between Centre for Ecology & Hydrology (CEH), the Met Office, National Oceanography Centre (NOC) and Plymouth Marine Laboratory (PML).

Part of this work has been carried out as part of the Copernicus Marine Environment Monitoring Service (CMEMS) Ocean-
15   Wave-Atmosphere Interactions in Regional Seas (OWAIRS) project. CMEMS is implemented by Mercator Ocean in the framework of a delegation agreement with the European Union.

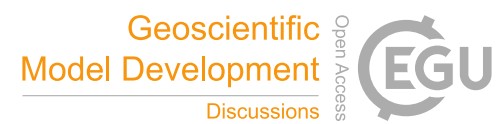

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





| | UKC2 | UKC3 |
|---|---|---|
| MetUM atmosphere code base | vn10.1 | vn10.6 |
| JULES land surface code base | vn4.2 | vn4.7 |
| Atmosphere/land science configuration | OS37 | RA1-M |
| NEMO ocean code base | vn3.6, r5518 | vn3.6, r6232 |
| Ocean science configuration | - | CO7 |
| WAVEWATCH III wave code | vn4.18 branch r1328 | vn4.18 branch r1782 |
| OASIS3-MCT coupling libraries | vn2.0 | vn2.0 |
| rose suite control tool | vn6.0 | vn2018.02.0 |
| Coupling science enabled[1] | atm ←→ ocn → wav ←→ atm | atm ←→ ocn ←→ wav ←→ atm |
| Model time step | 60 s | |
| Model domain | Rotated lat/lon coordinates, pole at actual position of 37.5° N, 177.5° E | |
| Simulation mode | Free running, no data assimilation | |
| Initialisation and boundary forcing | Operational atmosphere and ocean archives | |
| Coupling exchange frequency | Hourly, using hourly mean fields, and same frequency across all components | |
| Remap interpolation weights | Computed offline using ESMFregrid tool (Jones, 2015) | |
| Interpolation algorithm | First-order conservative for scalars, bilinear interpolation for vector fields | |

**Table 1: Summary of key differences and similarities between UKC3 configurations described in this paper and the preceding UKC2 system described by Lewis et al. (2018). [1] Note coupling science is described as being enabled between model X and Y in one-way as X --> Y, or two-way coupling modes as X <--> Y.**



| Configuration | Name | rose suite id[1] | RUNID | Description |
|---|---|---|---|---|
| Coupled | UKC3aow | u-ar588 | UKC3aow | *fully coupled* atmosphere-ocean-wave simulation |
| | UKC3ao | u-ar590 | UKC3ao | *'partially coupled'* atmosphere-ocean simulation |
| | UKC3aw | u-ar592 | UKC3aw | *'partially coupled'* atmosphere-wave simulation |
| | | | UKC3awf | as UKC3aw, with ocean forcing from external files |
| | UKC3ow | u-ar584 | UKC3owg | *'partially coupled'* ocean-wave, global meteorology |
| | | | UKC3owh | as UKC3owg, with high resolution meteorology forcing |
| | | | | |
| Atmosphere-only | UKA3 | u-ar585 | UKA3g | persisted OSTIA, global resolution SST lower boundary |
| | | | UKA3h | persisted 1.5 km resolution UKO3 SST lower boundary |
| | | | UKA3u | daily updated OSTIA at 1/20° resolution SST boundary |
| | | | | |
| Ocean-only | UKO3 | u-ar580 | UKO3g | global operational MetUM meteorological forcing |
| | | | UKO3h | high resolution UKA3 meteorological forcing |
| | | | UKO3gw | as UKO3g, with wave forcing from external files |
| | | | UKO3hw | as UKO3h, with wave forcing from external files |
| | | | | |
| Wave-only | UKW3 | u-ar583 | UKW3g | global operational MetUM wind forcing |
| | | | UKW3h | high resolution UKA3 wind forcing |
| | | | UKW3go | as UKW3g, with ocean forcing from external files |
| | | | UKW3ho | as UKW3h, with ocean forcing from external files |

**Table 2: Summary of UKC3 system coupled and uncoupled evaluation suites. [1] All configurations are available to registered researchers as rose suites via url links provided in the Table, from the https://code.metoffice.gov.uk/trac/roses-u/ repository. The various boundary condition and/or forcing options described can be enabled using the RUNID configuration parameter.**





| Order | Interface | Exchanged variable | Symbol | Units | Frequency | Time processing |
|---|---|---|---|---|---|---|
| 1 | W – A | Wave-dependent Charnock parameter | $\alpha$ | - | 1 hour | Hourly mean |
| | | | | | | |
| 2 | O – A | Sea surface temperature | $SST$ | K | 1 hour | Hourly mean |
| 2 | O – A | Zonal surface current | $u_{curr}$ | m s$^{-1}$ | 1 hour | Hourly mean |
| 2 | O – A | Meridional surface current | $v_{curr}$ | m s$^{-1}$ | 1 hour | Hourly mean |
| | | | | | | |
| 3 | O – W | Water level relative to local bathymetry | $D$ | m | 1 hour | Hourly mean |
| 3 | O – W | Zonal surface current | $u_{curr}$ | m s$^{-1}$ | 1 hour | Hourly mean |
| 3 | O – W | Meridional surface current | $v_{curr}$ | m s$^{-1}$ | 1 hour | Hourly mean |
| | | | | | | |
| 4 | A – O | Zonal wind stress on ocean surface | $\tau_x$ | N m$^{-2}$ | 1 hour | Hourly mean |
| 4 | A – O | Meridional wind stress on ocean surface | $\tau_y$ | N m$^{-2}$ | 1 hour | Hourly mean |
| 4 | A – O | Solar surface heat flux (all wavelengths) | $Q_{sr}$ | W m$^{-2}$ | 1 hour | Hourly mean |
| 4 | A – O | Non-solar net surface heat flux | $Q_{ns}$ | W m$^{-2}$ | 1 hour | Hourly mean |
| 4 | A – O | Rainfall rate | $R$ | kg m$^{-2}$ s$^{-1}$ | 1 hour | Hourly mean |
| 4 | A – O | Snowfall rate | $S$ | kg m$^{-2}$ s$^{-1}$ | 1 hour | Hourly mean |
| 4 | A – O | Evaporation of fresh water from ocean | $E$ | kg m$^{-2}$ s$^{-1}$ | 1 hour | Hourly mean |
| 4 | A – O | Wind speed at 10 m above ocean surface | $ws_{10}$ | m s$^{-1}$ | 1 hour | Hourly mean |
| 4 | A – O | Mean sea level pressure | $Pmsl$ | Pa | 1 hour | Hourly mean |
| | | | | | | |
| 5 | W – O | Significant wave height | $Hs$ | m | 1 hour | Hourly mean |
| 5 | W – O | Zonal Stokes drift velocity | $u_s$ | m s$^{-1}$ | 1 hour | Hourly mean |
| 5 | W – O | Meridional Stokes drift velocity | $v_s$ | m s$^{-1}$ | 1 hour | Hourly mean |
| 5 | W – O | Mean wave period | $T_{01}$ | s | 1 hour | Hourly mean |
| 5 | W – O | Fraction of atmospheric stress to ocean | $tauoc$ | - | 1 hour | Hourly mean |
| 5 | W – O | Wave-modified surface drag coefficient | $C_D$ | - | 1 hour | Hourly mean |
| | | | | | | |
| 6 | A – W | Zonal wind speed at 10 m above surface | $U_{10}$ | m s$^{-1}$ | 1 hour | Hourly mean |
| 6 | A – W | Meridional wind speed at 10 m height | $V_{10}$ | m s$^{-1}$ | 1 hour | Hourly mean |

**Table 3: Summary of coupling exchanges between atmosphere/land (A), ocean (O) and wave (W) components within the UKC3 regional coupled prediction system. Note that the W − O exchanges listed at order 5 are introduced for the first time in UKC3. Other variable coupling is as described by Lewis et al. (2018). Ensuring that exchanges occur between model components in the coupling order shown avoids system deadlocks within OASIS3-MCT. The coupling frequency highlights that all fields are currently exchanged every hour of the simulation time, and that all fields are computed as hourly mean values. See Sect. 2 for further details.**





|  | **UKA2, UKC2** | **UKA3, UKC3** |
|---|---|---|
| *Coupled and atmosphere-only mode configurations* | | |
| MetUM atmosphere model code base | vn10.1 | vn10.6 |
| Link to merged code copy repository | UKA2 | UKA3 |
| Dynamical core | ENDGAME[1] (Wood et al., 2014). | |
| Prognostic fields | three-dimensional wind components, virtual dry potential temperature, Exner pressure, dry density, mass mixing ratio of water vapour and cloud fields | |
| Model grid | Horizontal discretisation onto a regular grid with Arakawa C-grid staggering (Arakawa and Lamb, 1977) and a Charney-Phillips vertical staggering (Charney and Phillips, 1953) using terrain-following hybrid height coordinates | |
| Boundary layer scheme | First-order turbulence closure mixing adiabatically conserved heat and moisture variables, momentum and tracers as described by Lock et al. (2000) and Brown et al. (2008) | |
| Model resolution and domain | 950 cells across the west-east and 1025 cells in the north-south coordinate, based on variable resolution grid with inner region over UK and Ireland having horizontal resolution of 0.0135° (approximately 1.5 km at mid-latitudes) | |
| Vertical model levels | 70 vertical coordinates as used in the operational UKV implementation is used, with a terrain-following coordinate near the surface evolving to a constant height at 40 km above sea-level at the model top (16 levels defined in the lowest 1 km). See Lewis et al. (2018) Supplementary Material for details. Lowest model level for density is set at 2.5 m above the surface | |
| Initialisation | Reconfiguration from free-running simulation of global MetUM configuration | |
| Horizontal boundary conditions | Provided from free-running simulation of global MetUM configuration | |
| *Atmosphere-only UKA3 mode settings* | | |
| Persisted sea surface temperature lower boundary condition | UKA2g: OSTIA[2] interpolated onto global MetUM grid – fixed through run<br>UKA2h: SST from UKO2 simulation – fixed through run | UKA3g: as UKA2g<br>UKA3h: SST from UKO3 simulation<br>UKA3u: OSTIA on 1/20° native grid – updated daily (at 0000) through run |
| Surface currents boundary condition | Surface velocity assumed to be zero (i.e. no currents) | |
| Default Charnock parameter | 0.011 | |

**Table 4: Summary of UKA3 atmosphere component, and key similarities and differences to the UKA2 configuration described by Lewis et al. (2018). [1] Even Newer Dynamics for General Atmospheric Modelling of the Environment (Wood et al., 2014). [2] Operational Sea Surface Temperature and Sea Ice Analysis (Donlon et al., 2012). Direct links to merged code are provided to support collaboration with registered researchers. Further information on accessing the MetUM can be found at http://ww.metoffice.gov.uk/research/collaboration/um-partnership.**



| Namelist | MetUM namelist parameter | UKA2 (PS37) | UKA3 (RA1) | Comment |
|---|---|---|---|---|
| run_sl | monotone_scheme | 1,1,0,0,1 | 1,3,0,0,1 | PMF moisture conservation |
| run_bl | bl_res_inv | 0 | 1 | Spread entrainment fluxes at |
|  | l_new_kcloudtop | .false. | .true. | PBL top across inversion |
|  | l_reset_dec_thres | .false. | .true. | zone, and retune boundary |
|  | local_fa | 3 | 2 | layer mixing scheme |
| run_dyn | l_conservation_moist_zlf | .false. | .true. | Apply moisture conservation |
|  | zlf_conservation_moist_option | 1 | 2 | with ADAS, improving precip |
| run_precip | ndrop_surf | 7.5e+7 | 5.0e+7 | Reduce cloud droplet number |
|  | z_surf | 0.0 | 50.0 | to *ndrop_surf* at height *z_surf*. |
| run_calc_pmsl | l_pmsl_sor | .true. | .false. | More efficient Pmsl routine |
| run_stochastic | decorr_ts_pert_theta | - | 600.0 | Correlate stochastic boundary |
|  | i_pert_theta | 2 | 3 | layer perturbations of |
|  | i_pert_theta_type | 0 | 1 | temperature and moisture in |
|  | l_pert_shape | - | .true. | time to persist increments for |
|  | mag_pert_theta | 0.5 | 1.0 | longer. |
|  | z_pert_theta | 400.0 | 1500.0 |  |
| r2lwclnl | i_gas_overlap_lw | 6 | 4 | Improve treatment of gaseous |
|  | i_gas_overlap_lw2 | 6 | 4 | absorption, as described by |
|  | i_scatter_method_lw | 4 | 5 | Walters et al. (2017; Sect. |
|  | spectral_file_lw | 'sp_lw_ga3_1' | 'sp_lw_ga7' | 2.3) |
|  | spectral_file_lw2 | 'sp_lw_cloud3_0' | 'sp_lw_cloud7' |  |
| r2swclnl | i_gas_overlap_sw | 5 | 4 | Improve treatment of gaseous |
|  | i_gas_overlap_sw2 | 5 | 4 | absorption, as described by |
|  | l_ch4_sw | .false. | .true. | Walters et al. (2017; Sect. |
|  | l_n2o_sw | .false. | .true. | 2.3) |
|  | spectral_file_sw | 'sp_sw_ga3_0' | 'sp_sw_ga7' |  |
|  | spectral_file_sw2 | 'sp_sw_cloud3_0w' | 'sp_sw_cloud7' |  |

**Table 5: Summary of key changes between UKA2 and UKA3 configuration MetUM namelists, associated with implementing enhanced science options based on regional atmosphere-only model development and evaluation. Registered MetUM users can access further details at https://code.metoffice.gov.uk/trac/rmed/wiki/ra1/protoRA1.**





| | UKA2, UKC2 | UKA3, UKC3 |
|---|---|---|
| *Coupled and atmosphere-only mode configurations* | | |
| JULES land surface model code base | vn4.2 | vn4.7 |
| Link to merged code copy repository | UKL2 | UKL3 |
| Model resolution and domain | Model domain and horizontal resolution as atmosphere grid. | |
| Soil layers | 4 soil layers<br><br>Fixed layer thicknesses from the top down of 0.1 m, 0.25 m, 0.65 m and 2.0 m | |
| Surface tiling scheme | 9 surface tiles defined - five types of vegetation (broadleaf trees, needle-leaved trees, temperate $C_3$ grass, tropical $C_4$ grass and shrubs), four non-vegetated surface types (urban areas, inland water, bare soil and land ice), based on information from the Centre for Ecology & Hydrology Land Cover Map 2007 (CEH, 2007). | |
| Urban tile scheme | Best (2005) | MORUSES, Bohnenstengel (2011) |
| Soil hydraulic conductivity | Brooks-Corey following Cosby et al., (1984) | |
| Surface runoff generation | PDM, with optimised settings used as discussed by Martinez et al., (2018). | |
| River routing scheme | RFM kinematic wave equation (see Lewis et al., 2018 Appendix B for further details), parameter settings as described by Lewis et al., (2018) | |
| Soil moisture initialisation | Reconfiguration from free-running simulation of global configuration of MetUM | |
| River storage initialisation | Surface and sub-surface storage and grid cell inflow prognostics initialised from restart file of a multi-year standalone JULES simulation, driven by archived operational UKV NWP meteorological data. | |

**Table 6: Summary of UKA3 land surface component, and key similarities and differences to the UKA2 configuration described by Lewis et al. (2018). The direct links to merged code are provided to support collaboration with registered researchers. Further information on accessing JULES can be found at http://jules.jchmr.org.**





| Namelist | JULES parameter | UKL2 (PS37) | UKL3 (RA1) | Comment |
|---|---|---|---|---|
| jules_pftparm | alnir_io | 0.45,0.35,0.58,0.58,0.58 | 0.335,0.272,0.365,0.337,0.395 | Reduce amount |
| | alnirl_io | 0.30,0.23,0.39,0.39,0.39 | 0.30,0.23,0.30,0.30,0.30 | of bare soil in |
| | alpar_io | 0.10,0.07,0.10,0.10,0.10 | 0.073,0.041,0.090,0.106,0.074 | ancillary file and |
| | omega_io | 0.15,0.15,0.15,0.17,0.15 | 0.116,0.083,0.133,0.152,0.115 | change vegetated |
| | omegal_io | 0.10,0.10,0.10,0.12,0.10 | 0.10,0.05,0.10,0.12,0.10 | land tile scalar |
| | omnir_io | 0.70,0.45,0.83,0.83,0.83 | 0.818,0.544,0.738,0.683,0.785 | roughness and |
| | kext_io | 0.5,0.5,0.5,0.5,0.5 | 0.5,0.5,1.0,1.0,0.5 | albedo to improve |
| | z0hm_classic_pft_io | 1.65,1.65,0.10,0.10,0.10 | 1.65,1.65,0.01,0.01,0.10 | clear sky surface |
| | z0hm_pft_io | 1.65,1.65,0.10,0.10,0.10 | 1.65,1.65,0.01,0.01,0.10 | temperatures. |
| jules_radiation | l_niso_direct | .false. | .true. | As above |
| jules_snow | graupel_options | 0 | 1 | Avoid treating graupel as snow in JULES |

**Table 7: Summary of key changes between UKL2 and UKL3 configuration JULES namelists, associated with implementing enhanced science options based on regional atmosphere-only model development and evaluation. Registered MetUM users can access further details at https://code.metoffice.gov.uk/trac/rmed/wiki/ra1/protoRA1.**





| | UKO2, UKC2 | UKO3, UKC3 |
|---|---|---|
| *Coupled and ocean-only mode configurations* | | |
| NEMO ocean model code base | vn3.6, revision 5518 | vn3.6, revision 6232 |
| Link to merged code copy repository | UKO2 | UKO3 |
| Model domain and resolution | 1.5 km horizontal resolution, matching exactly where overlapping with inner domain of UKA2, requiring 1458 grid cells in the west-east zonal direction and 1345 grid cells in the north-south meridional direction, with Arakawa C-grid staggering (Arakawa and Lamb, 1977). | |
| Vertical levels | 51 vertical levels and a non-linear free surface. The vertical grid uses a stretched terrain following "S-coordinate" system as described by Siddorn and Furner (2013), | |
| Bathymetry | Based on EMODnet (EMODnet Portal, Sep 2015 release), using a minimum depth of 10 m, with no coastal wetting and drying imposed | |
| Eddy viscosity | For momentum and tracers, bilaplacian viscosities are applied on model levels (using coefficients of $6\times10^7$ $m^4s^{-1}$ and $1\times10^5$ $m^4s^{-1}$ respectively). | |
| Turbulence scheme | Generic Length Scale scheme is used to calculate turbulent viscosities and diffusivities (Umlauf and Burchard, 2003) and surface wave mixing is parameterised using the Craig and Banner (1994) scheme | |
| Bottom friction | Controlled through a log layer with a non-linear drag coefficient of 0.0025 | |
| Surface solar radiation | RGB light penetration scheme (see Lewis et al., 2018) for details; rn_abs=0.66 | |
| River discharge | Climatological river discharge data are applied as freshwater forcing (Graham et al., 2018) | |
| Initialisation | For case study simulations based on 2014 dates, initial conditions provided from a 1-year run of the AMM15 model initialised on 1 January 2014 from GloSea5 with meteorological forcing from the ERA-Interim reanalysis (Dee et al., 2011). For case study simulations based on 2015 dates, initial conditions are taken from a 1-year run of the UKO2 configuration initialised from the 2014 AMM15 hindcast on 1 January 2015. | |
| Horizontal boundary conditions | For case study simulations based on 2014 dates presented in Sect 5, daily boundary data of sea surface height, 2-d currents and 3-d temperature and salinity are provided from the archived ¼° resolution ocean data from the GloSea5 operational global seasonal forecast system (MacLachlan et al., 2015). For case study simulations based on 2015 dates in Sect 5, boundary data are provided from | |





| | the archived 12 km resolution NATL12 operational ocean model configuration (e.g. Siddorn et al., 2016). |
|---|---|
| Compilation keys[1] | key_zdfgls, key_dynspg_ts, key_ldfslp, key_vectopt_loop, key_bdy, key_tide, key_shelf, key_vvl, key_nosignedzero, key_iomput, *key_harm_ana,* key_netcdf4 |
| *Ocean-only or ocean-wave coupled mode configurations* | |
| Meteorological forcing | direct forcing approach, whereby the heat fluxes computed by an atmosphere model are applied, rather than being computed by NEMO based on bulk input properties. The *key_shelf* compilation key is also used, which implies that wind forcing is provided in the form of the $U$ and $V$ wind components rather than the surface stress components directly, and a surface layer parameterisation applied to translate to the stress forcing at the surface. <br> UKO2g and UKO3g – global MetUM operational forecast output <br> UKO2h and UKO3h – high resolution 1.5 km UKA2/UKA3 simulation output |

**Table 8: Summary of UKO3 ocean component, and key similarities and differences to the UKO2 configuration described by Lewis et al. (2018). Note that the NEMO compilation key *key_harm_ana* was only used in UKO3 implementations. [1] See Supplementary Material for description of compilation keys. The direct links to merged code are provided to support collaboration with registered researchers. Further information on accessing NEMO can be accessed at http://www.nemo-ocean.eu.**





| | UKW2, UKC2 | UKW3, UKC3 |
|---|---|---|
| *Coupled and wave-only mode configurations* | | |
| WAVEWATCH III model code base | vn4.18 | |
| WAVEWATCH III branch revision | r1328 | r1782 |
| Science configuration | As described by Lewis et al. (2018). By default apply "ST3" source terms (Komen et al., 1994) with tuning described by Bidlot et al. (2012). Nonlinear wave-wave nteractions parameterised using Discrete Interaction Approximation (Hasselmann et al., 1985). Wave enery propagation using second order upstream non-oscillatory scheme (Li, 2008) with 'Garden Sprinkler Effect' alleviation. | |
| Initialisation | Restart file generated by running the UKW* configuration from rest for the 5 day period prior to the case study initial time | |
| Boundary conditions | Spectral boundary conditions were provided from archived operational global wave model output, for which the WAVEWATCH III model resolution in open waters of the Atlantic was set at approximately 25 km. | |
| External forcing | UKW2g and UKW3g – operational global MetUM wind forcing only<br>UKW2h and UKW3g – high resolution 1.5 km UKA2/UKA3 wind forcing only<br>Forced wave-only simulations additionally including ocean current information read from file are termed UKW2c, with surface currents taken from UKO2h case study output. Finally, forced wave-only simulations termed UKW2l have also been run with wind, current and water level forcing, with the water levels also taken from the same UKO2h case study NEMO output. | |
| Compilation switches[1] | F90 MPI DIST OA3 NC4 NOGRB LRB4 ST3 STAB3 NL1 BT1 DB1 TR0 BS0 XX0 WNT1 WNX1 CRT1 CRX1 FLX0 LN1 RWND IC0 REF0 PR3 UNO RTD | |

**Table 9: Summary of UKW3 WAVEWATCH III wave model component, highlighting substantive aspects same as for UKW2 configuration described by Lewis et al. (2018). [1] A fuller description of the compilation switched is provided in the Supplementary Material. The direct links to merged code are provided to support collaboration with registered researchers. Further information on accessing WAVEWATCH III can be found at http://polar.ncep.noaa.gov/waves/wavewatch.**



|  | **Atmosphere/land** | **Ocean** | **Wave** |
|---|---|---|---|
| **Fully coupled** | UKC3aow | | |
| **Partially coupled** | UKC3ao | | UKC3owg |
| **Control (uncoupled)** | UKA3u | UKO3g | UKW3g |

**Table 10: Summary of model configurations used for system evaluation simulations relevant to each model component. See also Table 2 for configuration definitions.**

|  | **Coupled** | | | **Atmosphere** | **Ocean** | **Wave** |
|---|---|---|---|---|---|---|
| **Configuration** | UKC3aow | UKC3ao | UKC3owg | UKA3u | UKO3g | UKW3g |
| **Nodes used** | 40 | 29 | 18 | 48 | 15 | 11 |
| **Runtime / day** | 45 min | 50 min | 40 min | 30 min | 15 min | 30 min |
| **Output / day** | 100 Gb | 90 Gb | 60 Gb | 40 Gb | 50 Gb | 10 Gb |

5   **Table 11: Summary of computer resource usage and typical runtimes and volume of data outputs generated for each day of simulation of UKC3 systems. Note further optimisations of system node usage and run times are possible.**

|  | **UKO3gw or UKC3 configuration** |
|---|---|
| **[namelist:namsbc]** | ln_wave=.true. |
|  | nn_drag=1 |
| **[namelist:namsbc_wave]** | ln_sdw=.true. |
|  | ln_stcor=.true. |
|  | ln_cdgw=.true. |
|  | ln_tauoc=.true. |
|  | ln_phioc=.false. |
|  | ln_rough=.true. |
|  | nn_sdrift=1 |
| **[namelist:namzdf_gls]** | nn_z0_met=3 |
| **Variables read/coupled** |  |
| sn_usd | u-component Stokes drift |
| sn_vsd | v-component Stokes drift |
| sn_swh | Wave height |
| sn_wmp | Wave mean period |
| sn_tauoc | Momentum fraction to ocean |
| sn_cdg | Surface drag coefficient |

**Table B1: Summary of NEMO namelist configuration settings for enabling wave-to-ocean forcing or coupling.**



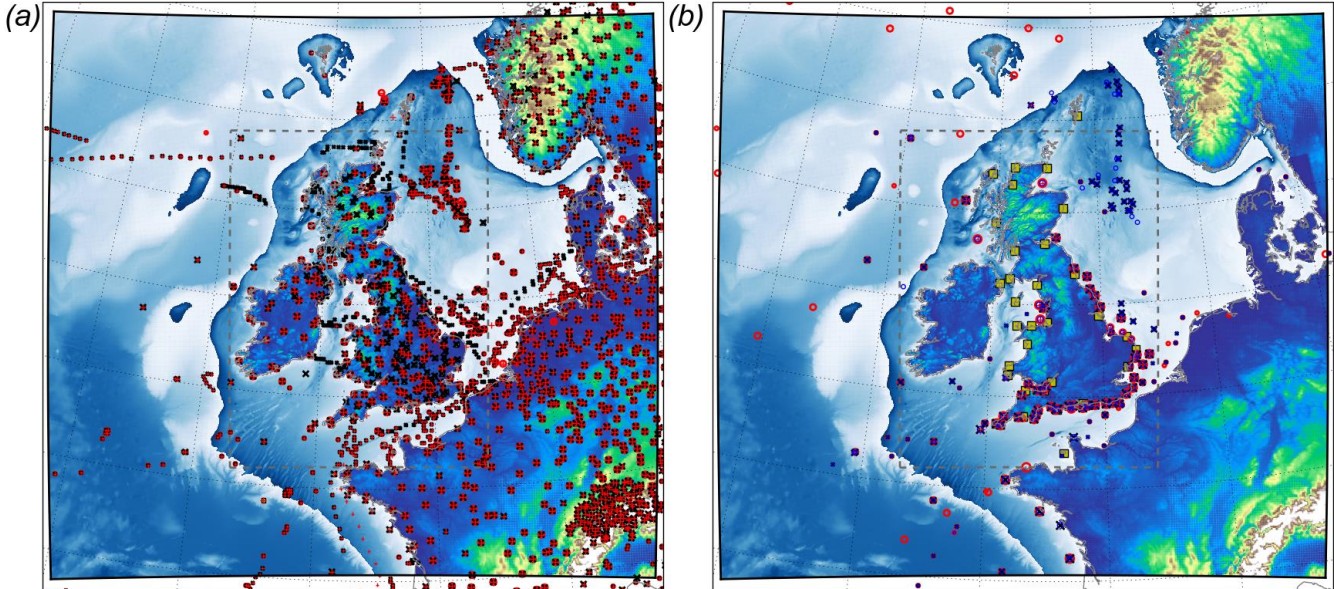

**Figure 1: Illustration of the UKC3 domain showing the extent of atmosphere/land model domain orography and ocean/wave model domain bathymetry. The regular 1.5 km resolution inner region of the atmosphere model grid is indicated by the gray dashed line. (a) Location of sample of in-situ observations on 15 July 2014, relevant for evaluating atmosphere model results. Key to symbols: red circle – visibility, black cross – air temperature, red cross – wind speed and direction. (b) Location of in-situ observations on 15 July 2014 relevant for evaluating ocean and wave components. Key: yellow squares – tide gauge sea surface height, red circle – sea surface temperature, black cross – maximum wave period, blue circle – significant wave height.**





**Figure 2: Wave model climatology from UKW3g of coupling-related variables – (a, d, g) normalised stress fraction tauoc, (b, e, h) wave-modified surface drag coefficient and (c, f, i) Stokes drift speed. The maps in (a, b, c) show monthly mean values from the July 2014 'summer' period and in (d, e, f) from the February 2015 'winter' period. Binned scatter plots on the bottom row show the simulated variation of (g) Charnock parameter, (h) drag coefficient and (i) Stokes drift speed as a function of wind speed across all simulated months in April, July, October 2014 and February 2015 at a point in the central North Sea. Similar distributions are found across the model domain. Colours show the frequency of data within each bin. In (h) the *nn_drag*=0 NEMO formulation is plotted by the S&B-75 blue dashed line.**





Figure 3: Sensitivity of ocean model Sea Surface Temperature to coupling. (a,d,j) Monthly mean difference between fully coupled and ocean only [UKC3aow – UKO3g] during April, July, October 2014 and February 2015 runs respectively. (b,e,h,k) Monthly mean difference between fully coupled and partially coupled [UKC3aow – UKC3ao] runs during each experiment. Note the different colour scale. (c,f,i,l) Percentage difference in Surface Temperature RMSE statistic for UKC3aow relative to UKO3g.





**Figure 4:** (a,c,e,g) Time series of average MODEL – OBSERVATION bias of ocean model sea surface temperature across all observing sites for each simulation for July 2014, April 2014, October 2014 and Feburary 2015 respectively. (b,d,f,h) Differences in absolute | MODEL – OBS | bias for each simulation period relative to UKA3u. A negative relative |average bias| indicates the coupled
5    system to have a lower average absolute bias across all observation sites than the control. Note that all plots have different scales across the different months evaluated.





**Figure 5: Sensitivity of wave model wind forcing, significant wave height (Hs) and mean wave period (T01) to coupling during October 2014 experiment. Monthly mean differences between UKC3aow and UKW3g for (a) coupled/forced atmosphere winds, (b) significant wave height Hs and (c) mean wave period T01. (d, e, f) Percentage difference in RMSE statistic for UKC3aow results relative to UKW3g for (d) wind forcing, (e) Hs, (f) T01. (g, h, i) Monthly mean differences between partially coupled UKC3owg and UKW3g (i.e. with same global-scale wind forcing) for (g) wind forcing, (h) Hs, (i) T01, and (j, k, l) Percentage difference in RMSE statistics for each variable for UKC3owg relative to UKW3g.**





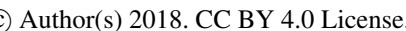

**Figure 6: Difference of wind forcing applied to wave model. (a,c,e,g) Time series of average MODEL – OBSERVATION bias across all observing sites for each simulation for July 2014, April 2014, October 2014 and Feburary 2015 respectively. (b,d,f,h)**
5  **Differences in absolute | MODEL – OBS | bias for each simulation period relative to the UKW3g wind forcing.**



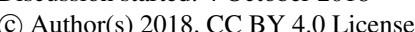



**Figure 7: Sensitivity of wave model Siginificant Wave Height (Hs) to coupling. (a,c,e,g) Time series of average MODEL –
OBSERVATION bias across all observing sites for each simulation for July 2014, April 2014, October 2014 and Feburary 2015
respectively. (b,d,f,h) Differences in absolute | MODEL – OBS | bias for each simulation period relative to UKW3g.**





**Figure 8: Sensitivity of wave model Wave Mean Period (T01) to coupling. (a,c,e,g) Time series of average MODEL – OBSERVATION bias across all observing sites for each simulation for July 2014, April 2014, October 2014 and Feburary 2015 respectively. (b,d,f,h) Differences in absolute | MODEL – OBS | bias for each simulation period relative to UKW3g.**





**Figure 9: Sensitivity of atmosphere model Surface Temperature to coupling. (a,d,j) Monthly mean difference between fully coupled and daily updated OSTIA [UKC3aow – UKA3u] during April, July, October 2014 and February 2015 runs respectively. (b,e,h,k) Monthly mean difference between fully coupled and partially coupled [UKC3aow – UKC3ao] runs during each experiment (c,f,i,l) Percentage difference in Surface Temperature RMSE statistic for UKC3aow relative to UKA3u.**



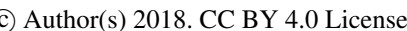


**Figure 10:** (a,c,e,g) Time series of average MODEL – OBSERVATION bias of surface temperature across all observing sites for each simulation for July 2014, April 2014, October 2014 and Feburary 2015 respectively. (b,d,f,h) Differences in absolute | MODEL – OBS | bias for each simulation period relative to UKA3u. A negative relative |average bias| indicates the coupled system to have a lower average absolute bias across all observation sites than the control.



**Figure 11: Sensitivity of atmosphere model Air Temperature at 1.5 m above surface to coupling. (a,d,j) Monthly mean difference between fully coupled and daily updated OSTIA [UKC3aow – UKA3u] during April, July, October 2014 and February 2015 runs respectively. (b,e,h,k) Monthly mean difference between fully coupled and partially coupled [UKC3aow – UKC3ao] runs during each experiment (c,f,i,l) Percentage difference in Air Temperature RMSE statistic for UKC3aow relative to UKA3u.**



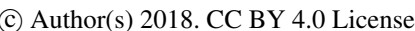


**Figure 12:** **(a,c,e,g) Time series of average MODEL – OBSERVATION bias of 1.5 m air temperature across all observing sites for each simulation for July 2014, April 2014, October 2014 and Feburary 2015 respectively. (b,d,f,h) Differences in absolute | MODEL – OBS | bias for each simulation period relative to UKA3u. A negative relative |average bias| indicates the coupled system to have a lower average absolute bias across all observation sites than the control.**





**Figure 13: Sensitivity of atmosphere model Wind Speed at 10 m above surface to coupling. (a,d,j) Monthly mean difference between fully coupled and daily updated OSTIA [UKC3aow – UKA3u] during April, July, October 2014 and February 2015 runs respectively. (b,e,h,k) Monthly mean difference between fully coupled and partially coupled [UKC3aow – UKC3ao] runs during each experiment (c,f,i,l) Percentage difference in wind speed RMSE statistic for UKC3aow relative to UKA3u.**





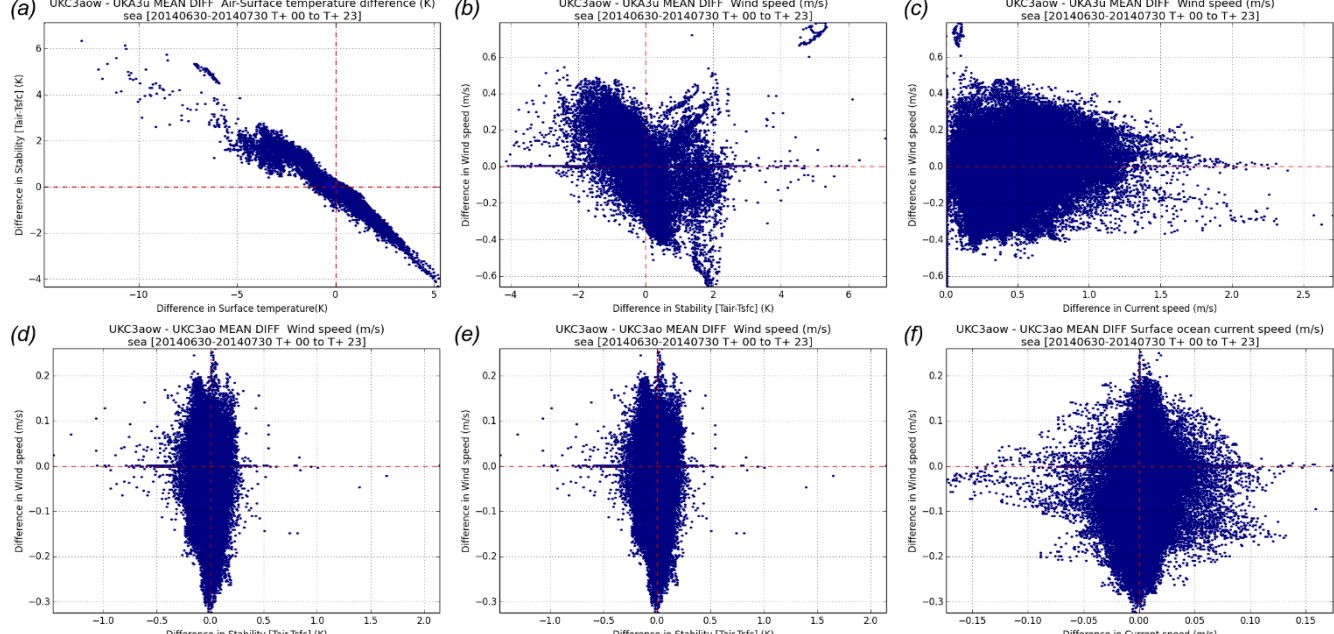

**Figure 14: Scatter plots showing relationships between differences in monthly mean results during July 2014 of (a, d) near-surface temperature difference (1.5 m air temperature – SST) with differences in SST, (b, e) 10 m wind speed with near-surface temperature difference and (c, f) 10 m wind speed with differences in surface current speed. Plots in (a, b, c) compare mean UKC3aow and UKA3u differences (i.e. fully coupled relative to atmosphere-only simulation). Plots in (d, e, f) compare mean UKC3aow and UKC3ao differences (i.e. fully coupled with wave relative to partially coupled atmosphere-ocean coupled).**





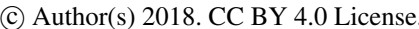

**Figure 15:** (a,c,e,g) Time series of average MODEL – OBSERVATION bias of 10 m wind speed across all observing sites for each simulation for July 2014, April 2014, October 2014 and Feburary 2015 respectively. (b,d,f,h) Differences in absolute | MODEL – OBS | bias for each simulation period relative to UKA3u. A negative relative |average bias| indicates the coupled system to have a lower average absolute bias across all observation sites than the control.