# Peer review of "The UKC3 regional coupled environmental prediction system"

_Geoscientific Model Development, 2018_

## Referee Comment (RC1) · Anonymous Referee #1 · 27 Oct 2018

General comments This paper describes the UKC3 system developed as part of the UK Environmental Prediction collaboration and how it has evolved from the previous UKC2. A rather substantive set of diagnostic results are presented. Notably, the impact of the increased resolution of the atmospheric forcing and the addition of the 2-way coupling to a model is presented. Results are very much encouraging and the authors have listed way to achieve further progress. The paper goes in quite some details of how the system can be set-up and at times, it reads quite like a manual rather than an scientific paper. Nevertheless, I still consider that such a description is quite valuable, in particular noting the very collaborative nature of this system development. As research is moving more and more in fully coupled system, this paper is a nice complement to research carried out elsewhere.

[Figure]

Specific comments p 13, line 13: could you elaborate a bit more why the NEMO turbulent kinetic energy budget due to wave processes are not included in UKC3? p 15, line 14: Are tau_wav and tau_wave:ocn computed from the wave model respective source term and if so, what was done for the contribution for frequencies above the last discretised frequency? The approach in Breivik et al. 2015 is to assume a balance between input and dissipation in the high frequency range. This is an assumption and truly speaking, it is not really correct as the nonlinear source term also contributes to flux of wave momentum and energy. Accounting for the nonlinear source term contribution and possible alternative methods to evaluate the momentum and energy fluxes are currently being investigated (Bidlot personal communication). p 15, line 29: the Stokes drift at other water depths than the surface could easily be computed. It has been deemed too expensive, hence the use of parameterisations to recover the Stokes drift profile. So I would change "known" to "usually available" Figure 5: Mean wave period reported by buoys tens to be based on the T02 (i.e. the second moment).According to the CEFAS WaveNet web page, they report "Average (zero crossing) wave period", which is T02. They also provide frequency spectra, so it is well possible to re-compute using any method. But then, one should make sure to use the same frequency range. Please clarify.

Technical corrections p16, lines 6 and 7: Phillips 2015 -> Breivik 2016 Appendix A: last entry: omega_p : units 1/s , name wave peak angular frequency p31, line 4: absorbed by the waves -> absorbed and/or released by the waves Table 9: wave-wave nteraction -> wave-wave interaction Figure 1: maximum wave period : do you mean Tp, the peak wave period ? Figures 2, 3, 9, 11, 13: (a,d,j) -> (a,d,g,j)

---

## Referee Comment (RC2) · Jagers (Referee) · 29 Dec 2018

General comments:

In the paper titled "The UKC3 regional coupled environmental prediction system" by Lewis et al, the researchers document the third generation of a regional coupled prediction system across sky, sea and land of the United Kingdom and the north-west European continental shelf. They start out by sketching the context in which their developments take place, both nationally and internationally. The system consists of an atmospheric component (tightly integrated with a land component), an ocean component and a wave component. The most important change from previous generation

UKC2 model is the option to pass the significant wave height, stokes drift velocity, mean wave period, reduction factor for atmospheric stress, and the wave-modified surface drag coefficient from the wave component to the ocean component to include the feedback from the waves on the ocean dynamics. The paper tries to follow a delicate balance between discussing these and other changes in detail and skipping over complexity in physics and implementation challenges. This discussion ends in a general discussion of simulation results; the presented results serve two aims: evaluating the effect of the new wave to ocean feedback (mostly a localized effect), and showing the model (stable) behavior over month-long simulations (compared to 5-day simulation for the UKC2 model in a previous paper). The combination of the discussion of technical details and complex simulations, could easily have been extended to a paper twice its current length. As a result the current paper is very interesting, but due to the high density of information and references at times also challenging to follow.

Specific comments:

1. Abstract, page 1, line 19-20: The major update is indicated to be "explicit representation of wave processes in the ocean and their feedbacks through wave-to-ocean coupling". This suggests that the wave component is new to the UKC3 system, but later in the paper it's indicated that the UKC2 model already included the waves component (forced by ocean currents and interacting with the atmosphere) and thus that the wave-to-ocean feedback coupling by including wave forces on the ocean are new.

2. Abstract, p1, l25: extended periods. The meaning of word "extended" only becomes clear on page 2, line 7/8 where it's indicated that periods are extended compared to the analysis of UKC2.

3. Sec 1.2, p3, l26-28. The term "component model" (or "component model technologies"?) is not defined. I've interpreted the text as: Developing increased

understanding and system improvements benefit from the application of a diversity of different simulation components and coupling technologies in a range of environments.

4. Sec 1.3, p6, l2. It's confusing that here UKA3 atmosphere and UKL3 land components are distinguished, whereas a couple of lines further in Sec 2, p6, l13 the combined atmosphere-land component is also indicated by UKA3. Throughout the paper UKA3 is used for both the atmosphere component and the combined atmosphere-land component (latter more frequent). Only Sec 2.2 discusses UKL3 as a separate component.

5. Sec 2.3, p12, l10-12. This is an important remark: the land fluxes don't run off into the ocean. So, the atmosphere-land and ocean models are not as fully coupled as may be suggested. This influences long term model stability.

6. Sec 4.2, p17, l26-28. No serious model drift found even without data assimilation. However, since the UKC3 is a local, nested model both the atmosphere and ocean models are significantly forced on these time scales by their boundary conditions. Furthermore, the land run off – a potential source for drift – is not connected to the ocean influx.

7. Sec 5, p25, l21-23. Data assimilation of observations in one part of the system may help to improve the state of coupled components as well, e.g. wave observations may help to improve atmospheric and oceanic state.

8. Sec 4.6, p23, l25-28. Very short section without any discussion about performance whereas in Sec 3, p13, l26-29 statements are included about the poor performance of coupled systems. The earlier remark demands a least a bit more discussion here. Possible effect of coupling frequency?

9. Table 2, p43.

- The Description column sometimes indicates that forcing comes from (external) files, sometimes just that the model is forced. The inconsistency in wording causes uncertainty about interpretation. Are the following additions correct?
- UKCao: no wave effects included
- UKCaw: no ocean currents included
- UKC3owg: global meteorology forcing from files
- UKA3g: no wave effects
- UKO3g/UKO3h: no wave forcing
- UKW3g/h: . . . forcing from files

10. Figure 2-15, p54-67, graphics.

- Text size is small and hard to read (same for Figs 3-15)

Technical corrections:

1. Abstract, p1, l25-27. Long sentence, consider breaking up or add a comma after "one month in duration" on line 25.

2. Abstract, p1, l28-29. The formulation that the results of the coupled model are "at least comparable skill to the equivalent uncoupled control simulations" suggest that the coupled approach does not show major improvements. Consider rephrasing to something like "The coupled approach shows notable improvements in surface temperature, wave state (in near-coastal regions) and wind speed over the sea, whereas the prediction quality of other quantities shows no significant improvement."

3. Sec 1.1, p2, l25. Reference Simpson (1992) not included unless Simpson (1997) is intended.

[Figure]

4. Sec 1.2, p3, l22. Reference Donelan et al. (2018) should read Donelan (2018)

5. Sec 1.2, p4, l7. Reference Skamarock et al. (2008) should read Skamarock and Klemp (2008).

6. Sec 1.2, p4, l26-27. For all the models and coupling techniques mentioned thus far references have been included, but not for NOGAPS atmospheric and NCOM ocean models mentioned here. This is inconsistent.

7. Sec 1.2, p4, l28  30. Reference Seo et al. (2017) should read Seo (2017); the reference Seo et al. (2007) is correct.

8. Sec 1.2, p5, l6-13. There are many other papers about the benefits of coupled wave-ocean models in coastal regions. This includes for instance:

   • Mulligan, R.P., Hay, A.R., Bowen, A.J., 2008. Wave-driven circulation in a coastal bay during the landfall of a hurricane, Journal of a Geophysical Research: Oceans, 113:C5, doi:10.1029/2007JC004500.

   • Uchiyama, Y., McWilliams, J.C., Shchepetkin, A.F., 2010. Wave-current interaction in an oceanic circulation model with a vortex-force formalism, Ocean Modelling, 34:1-2, 16-35, doi:10.1016/j.ocemod.2010.04.002.

   • Elias, E.P.L., Gelfenbaum, G., and Van der Westhuysen, A.J., 2012. Validation of a coupled wave-flow model in a high-energy setting: The mouth of the Columbia River, Journal of Geophysical Research: Oceans, 117:C9, doi:10.1029/2012JC008105.

9. Sec 1.3, p5, l24. Citing Martinez et al. (2018) as Martinez-de la Torre et al. (2018) is more consistent with other references such as "Luiz do Vale Silva et al. (2018)". Same in Sec 2.2, p10, l26 and Sec 2.2.1, p11, l7 and Sec 5, p26, l6 and Table 6.

10. Sec 2, p6, l28. Suggest to include the fully coupled configuration identification/RUNID: UKC3aow.

11. Sec 2, p7, l26. Reference Castillo et al. (2017) should read Castillo and Lewis (2017).

12. Sec 2.1, p8, l2, Remove duplicate period at end of line (after development..)

13. Sec 2.1, p8, l4. Suggest to include (PS37) after UKA2 to be consistent with UKA3 (RA1/RA1-M).

14. Sec 2.1, p8, l11-13. The phrase "in the context of the UK regional coupled prediction system" is irrelevant in the context of this paper; remove it for simplicity and clarity. Add "on this parameter" at the end of this sentence to put the "strong sensitivity" in context.

15. Sec 2.1, p8, l23. "A number of incremental updates have been introduced in the RA1-M science configuration". Only the pinned status of RA1-M is relevant not the way in which it was obtained . . . especially if not elaborated on further.

16. Sec 2.1, p8, l27 refers to a GA7 ticket. The introduction of Sec 2.1 indicates where the RA1 tickets can be found, but doesn't indicate what GA7 ticket numbers refer to.

17. Sec 2.1, p9, l3 and p10,l14 and Sec 2.2, p10, l14 refer to GA tickets. Are those GA7 tickets, or should the first ticket also refer to just GA?

18. Sec 2.1, p9, l7. The term PBL hasn't been defined. It probably refers to the planetary boundary layer, but this may not be clear for non-global non-atmospheric researchers.

[Figure]

19. Sec 2.1, p9, l11-14. Reference is made to specific values (1.0 K and 1500 m) while other sections state changes without reference to numbers. This seems inconsistent.

20. Sec 2.2, p10, l12. The reference CCL (2018) is missing.

21. Sec 2.4, p12, l29. The exact path is NOT indicated in Table 9.

22. Sec 3, p13, l19. No references included for POLCOMS and WAM; seems to be inconsistent.

23. Sec 3.1, p14, l31. Figure 2 should be replaced by Figures 2(b) and 2(e).

24. Sec 3.2, p15, l13. The use of tauoc suggests a correlation between all stresses with the local atmospheric stress. The $\tau_{wav:ocn}$ is unlikely to show such correlation at local scales (as resolution increases to resolving surf zones and estuaries).

25. Sec 3.3, p16, l8. Figure 2, plots (c), (f) and (i).

26. Sec 3.4, p16, l20-24. It would help readers if $\frac{665}{0.85}\left(\frac{c_p}{u_*}\right)^{3/2}\frac{u_*}{g}$ were introduced more clearly as an estimate for $H_s$.

27. Sec 4.2, p17, l21. Rather than "complements" consider "extends": This approach extends the analysis of Lewis et al. (2018) who considered only a number of relatively short 5-day case study simulations across a range of conditions to evaluate UKC2 performance.

28. Sec 4.2, p17, l24. That Lewis et al. (2018) didn't do long simulations doesn't imply that such simulations were never done, so remove "therefore" in the sentence "These experiments therefore represent the first time . . ."

29. Sec 4.2, p18, l10. "using more" instead of "usin more".

30. Sec 4.3, p18, l24-25. "In April, July and October runs . . . typically 0.2 K, but by up to 1 K during July 2014." July should probably not be included in the first list of months; should this be February?

31. Sec 4.3, p19, l18 refers incorrectly to Figure 4. This should be Figure 3.

32. Sec 4.3, p19, l9 l12 l13 l19 refer incorrectly to Figure 5. This should be Figure 4.

33. Sec 4.3, p19, l9. The list of subplots should probably include subplot 4(h).

34. Sec 4.4, p19, l32 refers incorrectly to Figure 6. This should be Figure 5.

35. Sec 4.4, p20, l15. Figure 5(c) should probably refer to Figure 5(b).

36. Sec 4.4, p21, l5. Reference Lewis et al. (2018a) doesn't exist.

37. Sec 4.4, p21, l7. "due to" instead of "du to"

38. Sec 4.5, p21, l16. Reference Donlon et al. (2008) doesn't exist.

39. Sec 4.5, p21, l27. "notable" instead of "nonable"

40. Sec 4.5, p21, l29. "coastline" instead of "coastaline"

41. Sec 4.5, p22, l20. "experiments" instead of "experimnets"

42. Sec 4.5, p22, l28. "relatively increased sea surface (and air) temperatures" instead of "relatively enhanced . . ."

43. Sec 4.5, p23, l1. "increased" instead of "increases"

44. Sec 5, p24, l2. "UKC3aow provides a truly coupled system" . . . still without water flowing from the land to the ocean (Sec 2.3, p12, l10-12), so not so truly coupled.

45. Sec 5, p24, l16. Most likely "... through further publications" instead of singular.

46. App B, p30, l5. "This capability is provided in NEMO from vn 4.0, ..." instead of "This capability is provided at from NEMO vn 4.0, ..."

47. App B, p30, l7. Which "Appendix III"?

48. App B.3, p32. This list of quantities that can be used during a wave to NEMO coupling includes three quantities "Normalized wave to ocean energy", "mean wave number" and "peak frequency" that are not actually used in UKC3. This seems to be inconsistent with the title and introduction of App B that indicate that these NEMO wave forcing changes were implemented for UKC3.

49. References, p34, l26. DOI seems to be completely incorrect, should read 10.1029/98JC02622.

50. References, p34, l29-30. Duplicate reference entry ... also p35, l1-3.

51. References, p35, l9. Bush et al (2018). Check names and submission status.

52. References, p37, l6. "Bakhoday Paskyabi" with space.

53. References, p37, l26. Kinter et al. (2012) not referred to.

54. References, p38, l3-4. Formatting deviates from rest of document.

55. References, p41, l18-23. Walters et al. (2017) in review. Check status.

56. Table 1, p42.

    - Most of the version information is also included in later tables. Consider restructuring to reduce duplication.

- The table lists the atmosphere/land configuration of UKC2 as OS37. Based on the rest of the paper and Lewis et al. (2018) this should probably be PS37.
- The table lists the atmosphere/land configuration of UKC3 as RA1-M. Throughout the paper seemingly random the terms RA1 and RA1-M are used. Consistency is suggested or clarify the difference.
- OASIS3-MCT coupling libraries are consistent across versions and should therefore be in a cell merged across columns.
- The "model domain" does not actually specify the model domain, but merely the model coordinates.

57. Table 4, p45.

- Why does this table not include the atmosphere/land configuration science configuration ids PS37 and RA1-M? Isn't this a key difference for the atmosphere component? The ids are included in Table 5 also about the UKA2/3 components.
- Wood et al. (2014) is not included in the references.
- Arakawa and Lamb (1977) is not included in the references. Also in Table 8.
- Charney and Phillips (1953) is not included in the references.
- Brown et al. (2008) is not included in the references.
- UKA3h simulation obtains SST from UKO3 simulation. Which simulation: UKO3g?

58. Table 6, p47.

- CEH (2007) is not included in the references.
- Best (2005) is not included in the references.

- PDM, RFM and UKV are not defined.

59. Table 8, p49-50.

  - Umlauf and Burchard (2003) is not included in the references.
  - Craig and Banner (1994) is not included in the references.
  - Dee et al. (2011) is not included in the references.
  - MacLachlan et al. (2015) is not included in the references.
  - Horizontal boundary conditions section refers to "simulations based on 2015 dates in Sect. 5". Reference doesn't seem to be correct. No relevant information found in Sec 5.
  - Siddorn et al. (2016) is not included in the references.

60. Tabl 9, p51.

  - WAVEWATCH III model code base are different (see Table 1).
  - Missing repository links as promised in Sec 2.4, p12, l29
  - Bidlot et al. (2012) is not included in the references.
  - Li (2008) is not included in the references.

61. Figure 2, p54, caption.

  - "(a, d, g) normalized stress fraction tauoc" – this list of subplots shouldn't include (g) which plots Charnock parameter.
  - "In (h) . . . SB-75 blue dashed line." There are multiple dashed lines. Remove unused and thus unnecessary lines.

---

## Author Response (AR1)

**Author Response to Reviewer RC1**

**The UKC3 regional coupled environmental prediction system**

Huw W. Lewis1, Juan Manuel Castillo Sanchez1, Alex Arnold1, Joachim Fallmann1,a, Andrew Saulter1, Jennifer Graham1,b, Mike Bush1, John Siddorn1, Tamzin Palmer1, Adrian Lock1, John Edwards1, Lucy Bricheno2, Alberto Martínez de la Torre3, James Clark4

**1** Response to general comments**

We would like to thank the reviewer for their thorough and complementary review. The list of specific comments provided is appreciated and these have been addressed in the revised manuscript, thereby improving the paper. We provide specific responses to these below.

**2 Response to specific comments**

a) p 13, line 13: could you elaborate a bit more why the NEMO turbulent kinetic energy budget due to wave processes are not included in UKC3?

Further clarification of this choice is now provided in the revised manuscript from p13, line 13. We focussed the initial implementation on improving the description of the momentum budget across atmosphere, ocean and wave components. Some tests were conducted using an implementation of wave effect in modifying the TKE budget (through the phioc parameter), but it is likely that other aspects of NEMO (e.g. mixing scheme) would require retuning in order to correct for compensating errors. This is therefore an aspect of ongoing work, with support from collaborations such as the NEMO Wave Working group.

b) p 15, line 14: Are tau\_wav and tau\_wave:ocn computed from the wave model respective source term and if so, what was done for the contribution for frequencies above the last discretised frequency? The approach in Breivik et al. 2015 is to assume a balance between input and dissipation in the high frequency range. This is an assumption and truly speaking, it is not really correct as the nonlinear source term also contributes to flux of wave momentum and energy. Accounting for the nonlinear source term contribution and possible alternative methods to evaluate the momentum and energy fluxes are currently being investigated (Bidlot personal communication).

In the version of WAVEWATCH III used in this work, the surface stress terms are only calculated in the model's numerical grid frequency range. No values are appended for the high frequency tail. As such, we interpret this to by similar to the approach in Breivik et al. 2015. A brief comment has been added to the revised manuscript to explicitly note this.

c) p 15, line 29: the Stokes drift at other water depths than the surface could easily be computed. It has been deemed too expensive, hence the use of parameterisations to recover the Stokes drift profile. So I would change "known" to "usually available"

We agree and have updated the manuscript in line with this suggestion.

d) Figure 5: Mean wave period reported by buoys tends to be based on the TO2 (i.e. the second moment). According to the CEFAS WaveNet web page, they report "Average (zero crossing) wave period", which is TO2. They also provide frequency spectra, so it is well possible to recompute using any method. But then, one should make sure to use the same frequency range. Please clarify.

The reviewer is correct that we should have provided comparisons of the observed mean wave period with T02 rather than T01 diagnostics from the wave model. As T02 was not readily available from all archived model simulations, the revised manuscript text and Figures 5 and 8 have been updated to discuss the wave peak period results, enabling comparison with more observation sites than the mean period (Fig. 1). The change of observed variable here does not impact our conclusions at all.

**3 Response to technical corrections identified**

p16, lines 6 and 7: Phillips 2015 -> Breivik 2016

This has been updated in the revised manuscript.

Appendix A: last entry: omega\_p : units 1/s , name wave peak angular frequency

Thank you for spotting this error – it has been corrected in the revised manuscript.

p31, line 4: absorbed by the waves -> absorbed and/or released by the waves

This has been corrected in the revised manuscript.

Table 9: wave-wave nteraction -> wave-wave interaction

This has been corrected in the revised manuscript.

*Figure 1: maximum wave period : do you mean Tp, the peak wave period ?* Yes, we mean the peak wave period. This has been corrected in the figure caption.

Figures 2, 3, 9, 11, 13: (a,d,j) -> (a,d,g,j)

This has been corrected in the revised manuscript where relevant for Figure captions 3, 9, 11, 13.

**Author Response to Reviewer RC2**

**The UKC3 regional coupled environmental prediction system**

Huw W. Lewis1, Juan Manuel Castillo Sanchez1, Alex Arnold1, Joachim Fallmann1,a, Andrew Saulter1, Jennifer Graham1,b, Mike Bush1, John Siddorn1, Tamzin Palmer1, Adrian Lock1, John Edwards1, Lucy Bricheno2, Alberto Martínez de la Torre3, James Clark4

**1** Response to general comments**

We would like to put on record our thanks to Dr Jagers for his very comprehensive and insightful review of this paper. It has clearly taken some considerable time, and we appreciate the efforts he has taken to support our contribution. We have taken account of all the comments raised and consider that these have greatly improved the revised manuscript. We provide specific responses to these below. The contribution of the reviewers has been acknowledged in the revised paper.

We note the reviewers' summary comment that "The combination of the discussion of technical details and complex simulations, could easily have been extended to a paper twice its current length. As a result the current paper is very interesting, but due to the high density of information and references at times also challenging to follow". We certainly sympathise with this perspective, noting that Reviewer 1 also commented that "The paper goes in quite some details of how the system can be set-up and at times, it reads quite like a manual rather than a scientific paper. Nevertheless, I still consider that such a description is quite valuable, in particular noting the very collaborative nature of this system development". We hope that this paper, particularly with the revisions made in response to the comments received, strike the balance between providing sufficient detail while ensuring readability. This is a particular challenge given the number of model components used within the system, but is increasingly common as prediction tools become more integrated. We have attempted to put much of the detail in the Appendices and summarising Tables 1-9. The revised manuscript has been checked again for readability, and some minor updates added. We also appreciate the reviewer's comment that there is much more that could be said even in relation to the simulations conducted to support this paper, but that this would lead to a much longer and unwieldy contribution! A number of other papers based on research using the UKC3 system (e.g. https://www.ocean-sci-discuss.net/os-2018-148/; https://www.ocean-sci-discuss.net/os-2018-162/ currently in open discussions) are currently in the review process, which will all cite the proposed GMD manuscript as an underpinning background reference.

**2 Response to specific comments**

 Abstract, page 1, line 19-20: The major update is indicated to be "explicit representation of wave processes in the ocean and their feedbacks through wave-to-ocean coupling". This suggests that the wave component is new to the UKC3 system, but later in the paper it's indicated that the UKC2 model already included the waves component (forced by ocean currents and interacting with the atmosphere) and thus that the wave-to-ocean feedback coupling by including wave forces on the ocean are new. The reviewer is correct. We had mean to imply that the wave effects in the ocean were new, rather than any representation of ocean waves, but can see this is not clear and this sentence has been revised as suggested.

2. Abstract, p1, l25: extended periods. The meaning of word "extended" only becomes clear on page 2, line 7/8 where it's indicated that periods are extended compared to the analysis of UKC2.

We agree, and have clarified this in the revised manuscript.

3. Sec 1.2, p3, I26-28. The term "component model" (or "component model technologies"?) is not defined. I've interpreted the text as: Developing increased understanding and system improvements benefit from the application of a diversity of different simulation components and coupling technologies in a range of environments.

This has been updated as suggested in the revised manuscript.

4. Sec 1.3, p6, l2. It's confusing that here UKA3 atmosphere and UKL3 land components are distinguished, whereas a couple of lines further in Sec 2, p6, l13 the combined atmosphere-land component is also indicated by UKA3. Throughout the paper UKA3 is used for both the atmosphere component and the combined atmosphere-land component (latter more frequent). Only Sec 2.2 discusses UKL3 as a separate component.

To improve clarity for the reader, we have opted to remove reference to "UKL3" as an explicit term, and made corrections where relevant to indicate "UKA3" to refer to the combined atmosphere-land system.

 Sec 2.3, p12, l10-12. This is an important remark: the land fluxes don't run off into the ocean. So, the atmosphere-land and ocean models are not as fully coupled as may be suggested. This influences long term model stability.

The reviewer is correct here. We plan to report more fully on developments to the land-ocean coupling and a more integrated representation of the water cycle in a subsequent documentation paper of a UKC4 system update. A line has been added to the manuscript to highlight this.

6. Sec4.2,p17,l26-28. No serious model drift found even without data assimilation. However, since the UKC3 is a local, nested model both the atmosphere and ocean models are significantly forced on these time scales by their boundary conditions. Furthermore, the land run off – a potential source for drift – is not connected to the ocean influx.

We note that the regional domain is relatively large in our system, such that model stability when run over extended periods is not guaranteed, but a further sentence has been added to the manuscript to reflect these external constraints on model stability in the regional system. 7. Sec 5, p25, l21-23. Data assimilation of observations in one part of the system may help to improve the state of coupled components as well, e.g. wave observations may help to improve atmospheric and oceanic state.

We agree, and note that initial simulations in an ocean-wave coupled mode only with ocean assimilation have now been attempted for this domain (https://www.ocean-sci-discuss.net/os-2018-148/). A line has been added to reflect this comment.

 Sec 4.6, p23, I25-28. Very short section without any discussion about performance whereas in Sec 3, p13, I26-29 statements are included about the poor performance of coupled systems. The earlier remark demands a least a bit more discussion here. Possible effect of coupling frequency?

We have extended Section 4.6 to provide some further commentary on the computational performance aspects. We have undertaken a number of coupling frequency experiments in the context of the UKC4 system, which we plan to report on in the subsequent documentation of that system rather than extend the scope of the current manuscript. In summary, we found the run times to be rather independent of the coupling frequency used (i.e. within the noise of run-to-run variability of run times, dependent on system load at time of run etc).

- 9. Table 2, p43. The Description column sometimes indicates that forcing comes from (external) files, sometimes just that the model is forced. The inconsistency in wording causes uncertainty about interpretation. Are the following additions correct?
  - UKCao: no wave effects included
  - UKCaw: no ocean currents included
  - UKC3owg: global meteorology forcing from files
  - UKA3g: no wave effects
  - UKO3g/UKO3h: no wave forcing
  - UKW3g/h: ... forcing from files

This is the correct interpretation, and we thank the reviewer for working through this. The descriptions in Table 2 have been revised to improve clarity for the reader.

**10. Figure 2-15, p54-67, graphics. Text size is small and hard to read (same for Figs 3-15)**

All figures have been replotted in the revised manuscript, including making all text larger and easier to read. Additional labels have also been included on the figure panels to better guide the reader as to what each is displaying. The new figures have replaced the original plots in the revised manuscript.

We also took the opportunity in reviewing and updating figures to provide updated plots which are based on a 'neighbourhood' comparison between model and observations. In this approach, a model mean of 3x3 grid cells nearest to each observation point are compared rather than the nearest grid point only. This is a first order attempt to provide a more robust comparison between model and observation at km-scale resolutions, in keeping with routine verification methods (e.g. Mittermaier et al., 2004). A line has been added to Sec. 4.2 of the revised manuscript to explain this. The change does not change any conclusions, but can be seen to lead to smoother summary time series plots for

example in Figs 4, 6, 7, 8, 10, 12, 15. We trust that all these changes contribute to a clearer presentation of results.

**3** Response to technical corrections**

1. Abstract, p1, I25-27. Long sentence, consider breaking up or add a comma after "one month in duration" on line 25.

This has been corrected.

2. Abstract, p1, I28-29. The formulation that the results of the coupled model are "at least comparable skill to the equivalent uncoupled control simulations" suggest that the coupled approach does not show major improvements. Consider rephrasing to something like "The coupled approach shows notable improvements in surface temperature, wave state (in near-coastal regions) and wind speed over the sea, whereas the prediction quality of other quantities shows no significant improvement."

Thank you for this helpful suggestion, now included in the revised manuscript.

3. Sec 1.1, p2, I25. Reference Simpson (1992) not included unless Simpson (1997) is intended.

Corrected.

4. Sec 1.2, p3, l22. Reference Donelan et al. (2018) should read Donelan (2018)

Corrected.

Corrected.

6. Sec 1.2, p4, l26-27. For all the models and coupling techniques mentioned thus far references have been included, but not for NOGAPS atmospheric and NCOM ocean models mentioned here. This is inconsistent.

Appropriate references (Bayler and Lewit, 1992; Barron et al., 2006) have been added.

7. 7. Sec 1.2, p4, l28 30. Reference Seo et al. (2017) should read Seo (2017); the reference Seo et al. (2007) is correct.

Corrected.

8. Sec 1.2, p5, l6-13. There are many other papers about the benefits of coupled wave-ocean models in coastal regions. This includes for instance:

Mulligan, R.P., Hay, A.R., Bowen, A.J., 2008. doi:10.1029/2007JC004500.
Uchiyama, Y., McWilliams, J.C., Shchepetkin, A.F., 2010, doi:10.1016/j.ocemod.2010.04.002.

• Elias, E.P.L., Gelfenbaum, G., and Van der Westhuysen, A.J., 2012. doi:10.1029/2012JC008105.

We thank the reviewer for these references, and agree that there are indeed many further besides. The intention in Sec 1.2 was to highlight most recent contributions (i.e. 2017, 2018 publications), which reflect updates to the literature since the UKC2 description paper was published in particular.

5. Sec 1.2, p4, I7. Reference Skamarock et al. (2008) should read Skamarock and Klemp (2008).

9. Sec 1.3, p5, l24. Citing Martinez et al. (2018) as Martinez-de la Torre et al. (2018) is more consistent with other references such as "Luiz do Vale Silva et al. (2018)". Same in Sec 2.2, p10, l26 and Sec 2.2.1, p11, l7 and Sec 5, p26, l6 and Table 6.

Corrected in all instances.

10. Sec 2, p6, I28. Suggest to include the fully coupled configuration identification/RUNID: UKC3aow.

This has been included in the bullet heading as suggested.

11. Sec 2, p7, l26. Reference Castillo et al. (2017) should read Castillo and Lewis (2017).

Corrected.

12. Sec 2.1, p8, l2, Remove duplicate period at end of line (after development..)

Corrected.

13. Sec2.1,p8,l4. Suggest to include (PS37) after UKA2 to be consistent with UKA3 (RA1/RA1-M).

Corrected.

14. Sec 2.1, p8, l11-13. The phrase "in the context of the UK regional coupled prediction system" is irrelevant in the context of this paper; remove it for simplicity and clarity. Add "on this parameter" at the end of this sentence to put the "strong sensitivity" in context.

The intention here was to highlight which changes, applied in RA1 with the driver of improving atmosphere-only simulations might be most important for the regional coupled simulations. The point being made here is that we find fog/cloud development to be sensitive to air-sea coupling, rather than making any particular comment on the sensitivity to the updated parameter. This sentence has been amended to clarify this point.

15. Sec 2.1, p8, l23. "A number of incremental updates have been introduced in the RA1-M science configuration". Only the pinned status of RA1-M is relevant not the way in which it was obtained ... especially if not elaborated on further.

Agreed. We have amended this sentence.

16. Sec2.1, p8, l27 refers to a GA7 ticket. The introduction of Sec2.1 indicates where the RA1 tickets can be found, but doesn't indicate what GA7 ticket numbers refer to.

We considered it sufficient to reference the GA7 documentation paper here, which includes sections labelled with the relevant GA7 ticket number and links to online documentation. This reference is therefore aiming to better sign-post interested readers to the relevant section of Walters et al. (2017). The aim of linking directly to the RA1 webpage, was in order to provide traceability to the atmosphere model configuration definition used.

17. Sec 2.1, p9, I3 and p10,I14 and Sec 2.2, p10, I14 refer to GA tickets. Are those GA7 tickets, or should the first ticket also refer to just GA?

These all refer to GA7 tickets, and the manuscript has been updated.

18. Sec 2.1, p9, I7. The term PBL hasn't been defined. It probably refers to the planetary boundary layer, but this may not be clear for non-global non-atmospheric researchers.

Agreed, this has been corrected.

19. Sec 2.1, p9, l11-14. Reference is made to specific values (1.0 K and 1500 m) while other sections state changes without reference to numbers. This seems inconsistent.

The change referred to here is relatively important for regional precipitation forecasting, and so was described here is a bit more detail, but we agree that this is inconsistent. Given the general comments on improving readability, we have removed reference to the specific values, and refer interested readers to the details provided by Bush et al., (2018) in the revised text.

20. Sec 2.2, p10, l12. The reference CCL (2018) is missing.

The reference here is to CCI (ESA Climate Change Initiative land cover), and the reference to https://www.esa-landcover-cci.org/ provided.

21. Sec 2.4, p12, l29. The exact path is NOT indicated in Table 9.

Table 9 has been updated to provide the relevant links.

22. Sec 3, p13, l19. No references included for POLCOMS and WAM; seems to be inconsistent.

References have now been included.

23. Sec 3.1, p14, l31. Figure 2 should be replaced by Figures 2(b) and 2(e).

Corrected.

24. Sec 3.2, p15, l13. The use of tauoc suggests a correlation between all stresses with the local atmospheric stress. The τwav:ocn is unlikely to show such correlation at local scales (as resolution increases to resolving surf zones and estuaries).

We agree with this comment, noting that this was the approach initially implemented by Breivik et al. (2015), but do not consider it relevant to add further detail to the manuscript here in interests of brevity. However we should note that the use of tauoc has been replaced by use of the wave model computed stress components directly within the UKC4 configuration (e.g. Lewis et al., 2018, <a href="https://www.ocean-sci-discuss.net/os-2018-148/">https://www.ocean-sci-discuss.net/os-2018-148/</a>), which should help correct this assumption. This would again need to be discussed in more detail in a subsequent publication documenting UKC4.

25. Sec 3.3, p16, l8. Figure 2, plots (c), (f) and (i).

Corrected.

26. Sec 3.4, p16, I20-24. It would help readers if <equation> were introduced more clearly as an estimate for Hs.

Agreed, and corrected in the revised manuscript.

27. Sec 4.2, p17, l21. Rather than "complements" consider "extends": This approach extends the analysis of Lewis et al. (2018) who considered only a number of relatively short 5-day case study simulations across a range of conditions to evaluate UKC2 performance.

Agreed. Updated in the revised manuscript.

28. Sec4.2,p17,l24. That Lewis et al. (2018) didn't do long simulations doesn't imply that such simulations were never done, so remove "therefore" in the sentence "These experiments therefore represent the first time ..."

Agreed. Updated in the revised manuscript.

29. Sec 4.2, p18, l10. "using more" instead of "usin more".

Corrected.

30. Sec 4.3, p18, l24-25. "In April, July and October runs ... typically 0.2 K, but by up to 1 K during July 2014." July should probably not be included in the first list of months; should this be February?

The intention was to suggest a mean relative cooling during those months due to coupling, with largest difference in July. This sentence has been rephrased to clarify in the revised text.

- 31. Sec 4.3, p19, l18 refers incorrectly to Figure 4. This should be Figure 3.
- 32. Sec 4.3, p19, l9 l12 l13 l19 refer incorrectly to Figure 5. This should be Figure 4.

The reviewer has correctly highlighted a number of Figure reference errors in Sec. 4.3. These have all been reviewed and corrected in the revised manuscript.

33. Sec 4.3, p19, I9. The list of subplots should probably include subplot 4(h).

Reference to Fig 4(h) is omitted as we highlight in particular April, July, October results where the improvement to SST due to coupling was found.

34. Sec 4.4, p19, l32 refers incorrectly to Figure 6. This should be Figure 5.

Corrected.

35. Sec 4.4, p20, l15. Figure 5(c) should probably refer to Figure 5(b).

The reviewer is right. This has been corrected.

36. Sec 4.4, p21, I5. Reference Lewis et al. (2018a) doesn't exist.

This has been corrected to Lewis et al. (2018).

37. Sec 4.4, p21, I7. "due to" instead of "dur to"

Corrected.

38. Sec 4.5, p21, l16. Reference Donlon et al. (2008) doesn't exist.

This should reference the Donlon et al. (2012) citation instead, and has been corrected.

39. Sec 4.5, p21, l27. "notable" instead of "nobable"

Corrected.

40. Sec 4.5, p21, l29. "coastline" instead of "coastaline"

Corrected.

41. Sec 4.5, p22, l20. "experiments" instead of "experimnets"

Corrected.

42. Sec 4.5, p22, l28. "relatively increased sea surface (and air) temperatures" instead of "relatively enhanced ..." Corrected.

43. Sec 4.5, p23, l1. "increased" instead of "increases"

Corrected.

44. Sec 5, p24, I2. "UKC3aow provides a truly coupled system" ... still without water flowing from the land to the ocean (Sec 2.3, p12, I10-12), so not so truly coupled.

The reviewer is correct. This sentence has been updated. We again note the intention to report on land-to-ocean coupling as one of the focus areas within the context of the subsequent UKC4 configuration development.

45. Sec 5, p24, l16. Most likely "... through further publications" instead of singular.

Corrected.

46. App B, p30, I5. "This capability is provided in NEMO from vn 4.0, ..." instead of "This capability is provided at from NEMO vn 4.0, ..."

Corrected.

47. App B, p30, I7. Which "Appendix III"?

Apologies, this has been corrected to read "Table B1", as a summary of potential NEMO wave coupling switches now supported.

48. App B.3, p32. This list of quantities that can be used during a wave to NEMO coupling includes three quantities "Normalized wave to ocean energy", "mean wave number" and "peak frequency" that are not actually used in UKC3. This seems to be inconsistent with the title and introduction of App B that indicate that these NEMO wave forcing changes were implemented for UKC3.

We agree that this is inconsistent. The first line of App B is true in that these changes were implemented as part of the development of UKC3, but we do not use all options available within the configuration. The title and introductory sentence of Appendix B have been amended.

49. References, p34, l26. DOI seems to be completely incorrect, should read 10.1029/98JC02622.

Corrected.

50. References, p34, I29-30. Duplicate reference entry ... also p35, I1-3.

Corrected.

51. References, p35, I9. Bush et al (2018). Check names and submission status.

A fuller reference with author list has been provided in the updated manuscript. We would anticipate this citation to be available in GMD Discussions prior to the review process for this paper being completed.

52. References, p37, I6. "Bakhoday Paskyabi" with space.

Corrected.

53. References, p37, l26. Kinter et al. (2012) not referred to.

Kinter is one of the co-authors of citation Jung et al. (2012), rather than a new citation.

54. References, p38, I3-4. Formatting deviates from rest of document.

Formatting of names corrected.

55. References, p41, l18-23. Walters et al. (2017) in review. Check status.

The status of the Walters et al. (2017) paper is unchanged at present. The citation is available at <a href="https://www.geosci-model-dev-discuss.net/gmd-2017-291/">https://www.geosci-model-dev-discuss.net/gmd-2017-291/</a>

**56. Table 1, p42.**

• Most of the version information is also included in later tables. Consider restructuring to reduce duplication.

• The table lists the atmosphere/land configuration of UKC2 as OS37. Based on the rest of the paper and Lewis et al. (2018) this should probably be PS37.

• The table lists the atmosphere/land configuration of UKC3 as RA1-M. Throughout the paper seemingly random the terms RA1 and RA1-M are used. Consistency is suggested or clarify the difference.

• OASIS3-MCT coupling libraries are consistent across versions and should therefore be in a cell merged across columns.

• The "model domain" does not actually specify the model domain, but merely the model coordinates.

Table 1 has been updated in line with these comments. The duplication has been maintained here in order that any reader only focussing on Table 1 has a simple reference to the key differences between UKC2 and UKC3, without needing to dig into the subsequent tables (i.e. it is hopefully clear which model components have changed versions).

57. Table 4, p45.

• Why does this table not include the atmosphere/land configuration science configuration ids PS37 and RA1-M? Isn't this a key difference for the atmosphere component? The ids are included in Table 5 also about the UKA2/3 components.

• Wood et al. (2014) is not included in the references.

- Arakawa and Lamb (1977) is not included in the references. Also in Table 8.
- Charney and Phillips (1953) is not included in the references.
- Brown et al. (2008) is not included in the references.
- UKA3h simulation obtains SST from UKO3 simulation. Which simulation: UKO3g?

Table 4 has been updated in line with these comments, and reference list updated with missing references. In practice, it is for a researcher to decide which high-resolution ocean simulation to use to initialise the UKA3h simulation. For example, one could now decide to use the SST from an operational AMM15 forecast. Typically the initial condition for UKO3g should also be the same as UKO3h. The ambiguity of referring to 'UKO3' is therefore perhaps appropriate here?

**58. Table 6, p47.**

- CEH (2007) is not included in the references.
- Best (2005) is not included in the references
- PDM, RFM and UKV are not defined.

Table 6 has been updated and the reference list updated in the revised manuscript.

**59. Table 8, p49-50.**

• Umlauf and Burchard (2003) is not included in the references.

- Craig and Banner (1994) is not included in the references.
- Dee et al. (2011) is not included in the references.
- MacLachlan et al. (2015) is not included in the references.
- Horizontal boundary conditions section refers to "simulations based on 2015 dates in Sect.
- 5". Reference doesn't seem to be correct. No relevant information found in Sec 5.
- Siddorn et al. (2016) is not included in the references.

Table 8 has been updated and the reference list updated in the revised manuscript.

60. Tabl 9, p51.

- WAVEWATCH III model code base are different (see Table 1).
- Missing repository links as promised in Sec 2.4, p12, l29
- Bidlot et al. (2012) is not included in the references.
- Li (2008) is not included in the references.

Table 9 has been updated in the revised manuscript. The WAVEWATCH III code base in UKC2 and UKC3 are the same (i.e. vn4.18), but different code branch revisions are used, as reflected in row 4 of Table 9. The missing repository links have now been added. Missing references have also been included.

61. Figure 2, p54, caption.

• "(a, d, g) normalized stress fraction tauoc" – this list of subplots shouldn't include (g) which plots Charnock parameter.

• "In(h) ... SB-75 blue dashed line." There are multiple dashed lines. Remove unused and thus unnecessary lines.

Figure 2 caption has been amended, and figure (h) simplified to include only the S&B dashed line described in the text.

**The UKC3 regional coupled environmental prediction system**

Huw W. Lewis1, Juan Manuel Castillo Sanchez1, Alex Arnold1, Joachim Fallmann1,a, Andrew Saulter1, Jennifer Graham1,b, Mike Bush1, John Siddorn1, Tamzin Palmer1, Adrian Lock1, John Edwards1, Lucy Bricheno2, Alberto Martínez de la Torre3, James Clark4

1Met Office, Exeter, EX1 3PB, UK
 2National Oceanography Centre, Liverpool, L3 5DA, UK
 3Centre for Ecology & Hydrology, Wallingford, OX10 8BB, UK
 4Plymouth Marine Laboratory, Plymouth, PL1 2LP, UK

[revised manuscript text omitted]

Note that, as in Breivik et al. (2015),  $\tau_{wav}$  and  $\tau_{wav:ocn}$  terms are computed from the wave model source terms across the model's frequency range only. This implies that input and dissipation are balanced at higher frequencies, with further work required to fully account for the tails of the wave frequency range.

30 Figure 2(a) and Figure 2(d) show the mean simulated tauoc for a summer and winter month, and highlights values tend to lie in the range 0.95 to 1.05 (i.e. order 5% modification to the atmosphere surface stress due to waves). Largest enhancement can be found along west-facing coastlines, and largest reductions in the lee of land such as downstream of the Scottish islands, in the Irish Sea and along the English Channel. The spatial distribution is broadly consistent between summer and winter months, but with the magnitude of wave modification clearly increased in winter.

**3.3 Stokes-Coriolis drift**

The Stokes drift, caused by finite amplitude waves, creates a relative motion along the wave direction which quickly decays

5 with depth. The NEMO momentum equation is modified to account for the Stokes drift velocity  $v_s$ , taking into account the Coriolis forcing, as in Eq. (4).

$$\frac{D\boldsymbol{u}}{Dt} = -\frac{1}{\rho_w} \nabla p + (\boldsymbol{u} + \boldsymbol{v}_s) \times f \hat{\boldsymbol{z}} + \frac{1}{\rho_w} \frac{d\tau}{dz},$$
(4)

As only the surface Stokes drift,  $v_0$ , is usually availableknown from the wave model, different parameterizations are used to estimate the change in the Stokes drift velocity with depth,  $v_s(z)$ , as a function of the mean wave period,  $t_{01}$ , significant wave

10 height  $H_s$ , and peak wave frequency  $\omega_p$ . Options are controlled by the *nn\_sdrift* NEMO namelist parameter. For *nn\_sdrift=0*, the Breivik 2015 parameterization is used (Breivik et al., 2015; (Eq. 120)), with the Stokes drift velocity profile  $v_s(z)$  given by Eq. (5). If *nn\_sdrift=1*, the Phillips parameterization (Breivik et al., 2016 (Eq. 100)) is applied using an inverse depth scale, according to Eq. (6). An extension can be applied if *nn\_sdrift=2* using the peak wave number as calculated by the wave model rather than the inverse depth scale, as shown in Eq. (7).

15 0:
$$v_{s}(z) = v_{0} \frac{e^{2k_{e}z}}{1-8k_{e}z}$$
  $k_{e} = \frac{|v_{0}|}{5.97} \frac{16}{2\pi} \frac{t_{01}}{H_{s}^{2}}$  (Breivik et al., 2015), (5)
1:  $v_{s}(z) = v_{0} \left[ e^{2k_{e}z} - \beta \sqrt{-2k_{e}\pi z} \operatorname{erfc}(\sqrt{2k_{e}z}) \right]$   $k_{e} = \frac{|v_{0}|}{5.97} \frac{16}{2\pi} \frac{t_{01}}{H_{s}^{2}}$  (Breivik et al., 2016Phillips 2015), (6)
2:  $v_{s}(z) = v_{0} \left[ e^{2k_{e}z} - \beta \sqrt{-2k_{p}\pi z} \operatorname{erfc}(\sqrt{2k_{p}z}) \right]$   $k_{p} = \frac{\omega_{p}^{2}}{g}$  (Breivik et al., 2016Phillips 2015), (20)

[revised manuscript text omitted]